# Orange-spotted grouper nervous necrosis virus-encoded protein A induces interferon expression via RIG-I/MDA5-MAVS-TBK1-IRF3 signaling in fish cells

Siyou Huang,[1] Yi Huang,[1] Taowen Su,[1] Runqing Huang,[2] Lianpan Su,[1] Yujia Wu,[1] Shaoping Weng,[1] Jianguo He,[1] Junfeng Xie[1]

**ABSTRACT**   Nervous necrosis virus (NNV), a highly contagious fish virus, has caused huge economic losses to the global aquaculture industry. A previous study showed that protein A (ProA) encoded by orange-spotted grouper NNV triggers type I interferon (IFN) production in fish cells, but the activation and modulation of correlative signal pathways remain unclear. Here, we proved that ProA induces fish cell-specific IFN promoter activation in a dose-dependent manner. In channel catfish ovary (CCO, an NNV-permissive cell), ProA evoked expression and secretion of functional IFN with anti-DNA virus activity, suggesting a good model for IFN signaling research. A contrastive study of signaling in CCO and fathead minnow (FHM, an NNV-nonpermissive cell) using RNAi knockdown or dominant negative mutant overexpression verified that ProA-mediated IFN activation went through retinoic acid-inducible gene I (RIG-I)-like receptor (RLR) pathway (RIG-I/MDA5-MAVS-TRAF3-TBK1-IRF3) while NOD1-RIPK2-NFκB and TLR3-TRIF branches were unnecessary. As RNA receptors upregulated by ProA in FHM, RIG-I and MDA5 promoted ProA-mediated IFN activation to form a positive feedback loop, while LGP2, NOD1, and PKR inhibited this activation as negative modulators. In ProA-expressed, NNV-infected CCO, the transcription of RIG-I, MDA5, MAVS, TBK1, IRF3, and IFN was downregulated, but the expression of LGP2, TRAF3, and NOD1 remained unchanged, suggesting unknown IFN suppression mechanism by NNV infection. IFN inhibition by overexpressing mutated RIG-I greatly enhanced NNV replication in FHM, implying that RIG-I might be the main target for both ProA-mediated activation and NNV infection-induced inhibition. This study provides overviews and foundations for understanding the interaction between betanodavirus-encoded protein and fish innate immune signaling.

**IMPORTANCE**   As a major pathogen, nervous necrosis virus (NNV) infects more than 120 fish species worldwide and is virulent to larvae and juvenile fish, hampering the development of the fish fry industry. Understanding virus-host interaction and underlying mechanisms is an important but largely unknown issue in fish virus studies. Here, using channel catfish ovary and fathead minnow cells as models for the study of innate immunity signaling, we found that NNV-encoded ProA activated interferon signaling via the retinoic acid-inducible gene I (RIG-I)-like receptor (RLR) pathway which was still suppressed by the infection of wild-type NNV. This finding has important implications for the comprehension of NNV protein function and the immune response from different cells. First, RIG-I is the key node for anti-NNV innate immunity. Second, the response intensity of RLR signaling determines the degree of NNV proliferation. This study expands our knowledge regarding the overview of signal pathways affected by NNV-encoded protein and also highlights potential directions for the control of aquatic viruses.

Address correspondence to Junfeng Xie, xiejf@mail.sysu.edu.cn, or Jianguo He, lsshjg@mail.sysu.edu.cn.

Yi Huang and Taowen Su contributed equally to this article. Author order was determined alphabetically.

The authors declare no conflict of interest.

See the funding table on p. 20.

**KEYWORDS** nervous necrosis virus, protein A, channel catfish, interferons, RLR signaling, RIG-I

The innate immune system is the first line of defense against invasive pathogens which utilizes a set of pathogen recognition receptors (PRRs) to identify pathogen components, called pathogen-associated molecular patterns (1). There are several PRR families that have been identified to be involved in the detection of viral-derived nucleic acids; retinoic acid-inducible gene I (RIG-I)-like receptors (RLRs) including RIG-I, MDA5, and LGP2; Toll-like receptors (TLRs), nucleotide oligomerization domain (NOD)-like receptors (NLRs); protein kinase R (PKR) (2–4); the DNA sensor cyclic guanosine monophosphate–adenosine monophosphate synthase (cGAS) (5), and so on. RLRs can detect double-stranded RNA (dsRNA) and single-stranded RNA (ssRNA) with 5′ppp ends (6, 7). Several TLRs, such as TLR3/7/8, also can recognize RNA (8–10). NOD1 and NOD2 have been considered sensors of peptidoglycan constituents of bacteria (11). However, it has been reported that NOD2 is able to recognize ssRNA to trigger the activation of IRF3 and the production of interferon-β (12). NOD1 has also been reported to detect dsRNA and stimulate an innate immune response (13). After detecting viral-derived nucleic acids, the cytosolic viral sensors recruit two adaptors, the mitochondrial antiviral signaling protein (MAVS) (14) and the stimulator of interferon genes (STING) (15, 16), to induce the production of type I interferons (IFNs) via activating tumor necrosis factor receptor-associated factor 3 (TRAF3), TANK-binding kinase 1 (TBK1), and IFN regulatory factor 3 (IRF3) axis (17) and subsequently to trigger a number of interferon-stimulated genes (ISGs) (18) via JAK-STAT signaling (19), exerting diverse functions at multiple levels of antiviral immunity (20).

Nervous necrosis virus (NNV) belongs to the family Nodaviridae genus *Betanodavirus* and is one of the most harmful fish viruses, leading to almost 100% mortalities in juveniles and larvae when infected. It is a small naked RNA virus containing two particles of positive single-stranded RNAs (RNA1 and RNA2) genome (21) within a 28-nm icosahedral viral capsid (22). There are four proteins, including protein A (ProA), capsid protein (CP), B2, and B1, encoded by these two RNA segments during the life cycle (23). The only structural protein, CP, had been developed as a virus-like particle vaccine (24, 25), proved to induce incomplete autophagy (26) and negatively regulate host type I IFN production (27), indicating that CP is a multifunctional viral protein. ProA acting as RNA-dependent RNA polymerase (RdRp) is responsible for replicating the viral genome on the outer membrane of mitochondria (28, 29). Our previous study proved that ProA can induce IFN expression in fathead minnow (FHM) cells, depending on mitochondrial localization and RdRp catalytic activity (30). In addition, activating of interferon-inducible factor 3 (IRF3) has been demonstrated to be essential for IFN induction by ProA. All results suggested that ProA induces IFN expression depending on RNA synthesis and cytoplastic nucleic acid receptors. However, this previous outcome of ProA-mediated IFN expression was studied in FHM, an NNV susceptible but nonpermissive cell line. Therefore, NNV-susceptible and NNV-permissive cells with the ability of IFN production should be used to elucidate the authentic role of ProA in the RLR signal pathway.

To uncover the detailed mechanism of ProA-mediated IFN production, we first used the orange-spotted grouper nervous necrosis virus (OGNNV) and channel catfish ovary (CCO) cells to set up an NNV-CCO infection model and confirmed that ProA maintained the anti-Iridovirus function in this model. By the comparative study of the related signal pathway in CCO and FHM, we roughly elucidated the related signaling axis needed for ProA-mediated IFN activation, identified the necessary factors and negative regulators, and showed the innate immune response under NNV infection. The overall modulations of ProA-mediated, IFN-related signaling in CCO and FHM cells were revealed, providing foundations for understanding the interaction between NNV-encoded proteins and host innate immune signaling.

## RESULTS

### ProA activates piscine IFN promoter activities in a dose-dependent manner

To verify the universality of IFN activation induced by NNV-encoded ProA, luciferase reporter assays were conducted by transiently transfecting several piscine cells and human cells with increasing amounts of ProA expression plasmid and corresponding IFN-stimulated response element (ISRE) reporter plasmids. The results showed that overexpression of ProA induced piscine IFN promoter activities in a dose-dependent manner in FHM, CCO, Asian sea bass (SB), and zebrafish embryonic fibroblast (ZF4) cells (Fig. 1A), indicating a universal characteristic of IFN activation. Although the IFN promoters were activated in all the tested fish cells, the response intensity was different. The most robust IFN response in the range of 10- to 120-fold was observed in FHM cells, while the weakest one below 1.5-fold was found in SB cells. CCO and ZF4 were in the middle range within 10-fold. However, the activation from ProA could not be repeated in human cells including 293T, HeLa, and HepG2 (Fig. 1B). ProA expressed well in human cells (Fig. 1A), but it could not induce the IFN activation whether at 37°C, the optimal temperature for cells, or 28 °C, the suitable temperature for ProA (Fig. S1B).

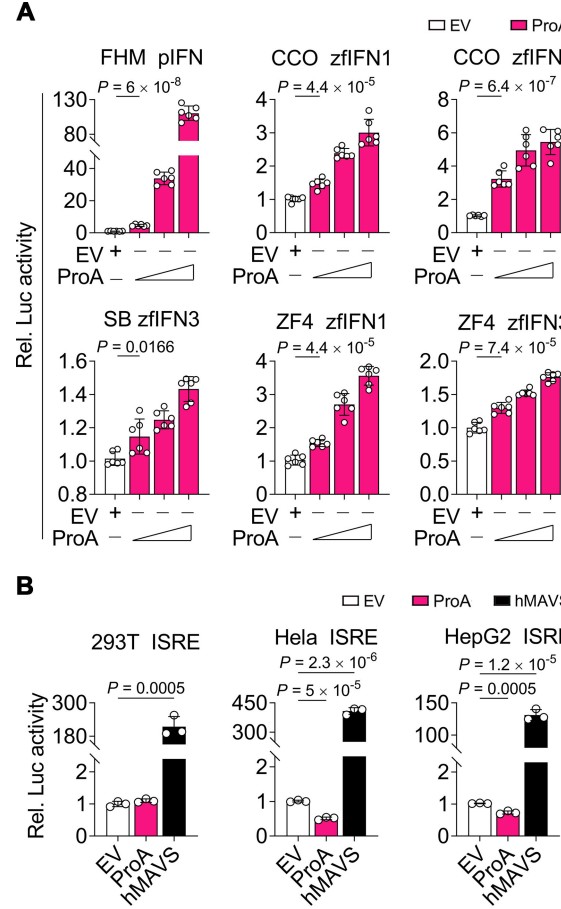

**FIG 1** ProA induces species-specific IFN promoter activation in a dose-dependent manner. (A and B) Luciferase activity induced by ProA. ProA plasmid was transfected into four kinds of piscine cells (A) and three kinds of human cells (B) with indicated luciferase reporter plasmids to perform a dual-luciferase reporter assay. Different IFN promoters were shown as follows: pIFN = FHM, zfIFN1/3 = zebrafish, and ISRE = human. Empty vector (EV) served as the negative control in all cells, and human MAVS (hMAVS) was used as a positive control in human cells. For A, experiments were repeated at least six times ($n = 6$) independently with similar results obtained, while $n = 3$ for B. Means ± SD are shown, and $P$ values were calculated using two-tailed unpaired Student's $t$-test.

## ProA enhances the anti-TFV (tiger frog virus) activity in CCO cells by upregulating IFN expression and secretion

Although CCO expresses a low level of IFN and ISGs (Fig. S2A), it can be infected by NNV (Fig. S2B) and TFV. Overexpression of ProA does not harm CCO cells (Fig. S2C). TFV-CCO infection model was used to test the ProA-induced antiviral function of IFN. ProA-overexpressed CCO cells could also show anti-TFV activity (Fig. S2D) in a dose-dependent manner (Fig. S2E; Table 1). Therefore, CCO is an ideal cell line for ProA-mediated IFN activation study. Because of the shortage of species-specific antibodies for these two cells, we monitored three markers of ProA-triggered IFN pathway activation mainly by promoter activation, transcript production, and IFN function of anti-TFV activity. Crystal violet staining of infected monolayer cells and virus titer determination (Table 1) indicated that ProA-transfected CCO cells showed resistance to TFV infection (Fig. 2A). This antiviral activity was generated from the functional ProA by upregulating the expression of IFN and downstream ISGs, including Mx1, PKR, and Viperin (Fig. 2B). It was further proved that ProA did not affect TFV entry (Fig. 2C, left bar chart) by determining the same *mcp* gene quantities in both EV- and ProA-transfected samples at a short period of 2 h but strongly inhibited viral replication (Fig. 2C) by evaluating the significant amount change of *mcp* after the long-term infection of 2 d post infection (dpi). Moreover, the supernatant from ProA-expressed cells exhibited similar anti-TFV activity (Fig. 2D) and ISG upregulation (Fig. 2E), even though the exosome was removed (Fig. 2F), suggesting that the secreting IFN is functional and not related to exosome. From the assays mentioned above, the ProA-mediated anti-TFV activity in CCO is dependent on its mitochondrial localization and RdRp activity because overexpression of mitochondrial targeting sequence-omitted mutant (ΔTM), RdRp domain-deleted mutant (ΔR), and RdRp catalytic activity motif mutant (GDD/AAA) lost the anti-TFV activity and IFN/ISG upregulation as the same level of EV control.

## RIG-I and MDA5 are essential for ProA-mediated IFN activation

FHM, NNV-nonpermissive cells, and CCO, NNV-permissive cells (Fig. S2B), were good materials for comparing the immune signaling and were selected for further study for

**TABLE 1** TFV titer ($TCID_{50}$) in different treated samples from indicated figures

| Sample[a] | Test1 | Test2 | Test3 | Average | P (to EV)[b] |
|---|---|---|---|---|---|
| Fig. S2E | | | | | |
| EV | 6.845 | 6.825 | 6.725 | 6.798 | |
| ProA-0.1 | 5.545 | 5.5 | 5.575 | 5.54 | 8.15E-06 |
| ProA-0.5 | 4.015 | 3.975 | 4.125 | 4.038 | 1.18E-06 |
| ProA-1 | 3.785 | 3.725 | 3.8 | 3.77 | 2.58E-07 |
| **Sample** | **Test1** | **Test2** | **Test3** | **Average** | **P (to ProA)** |
| Fig. 2A | | | | | |
| EV | 5.73 | 5.75 | 5.74 | 5.74 | 1.78E-05 |
| ProA | 4.125 | 4 | 3.875 | 4 | |
| △TM | 5.875 | 5.375 | 5.625 | 5.625 | 0.000547 |
| △R | 5.5 | 5.625 | 5.125 | 5.417 | 0.001051 |
| GDD/AAA | 5.875 | 5.75 | 5.625 | 5.75 | 6.79E-05 |
| Fig. 2D | | | | | |
| EV | 5.875 | 6.375 | 6 | 6.083 | 0.000359 |
| ProA | 4.375 | 4.375 | 4.25 | 4.333 | |
| △TM | 6.125 | 5.875 | 6 | 6 | 3.69E-05 |
| △R | 5.625 | 6.125 | 5.875 | 5.875 | 0.000508 |
| GDD/AAA | 5.875 | 5.75 | 5.625 | 5.75 | 7.02E-05 |

[a]The name of the corresponding plasmid-transfected sample was indicated. The number represented the transfecting amount in micrograms.
[b]The P values were calculated by two-tailed unpaired Student's *t*-test. The group as the comparative target was indicated in the bracket.

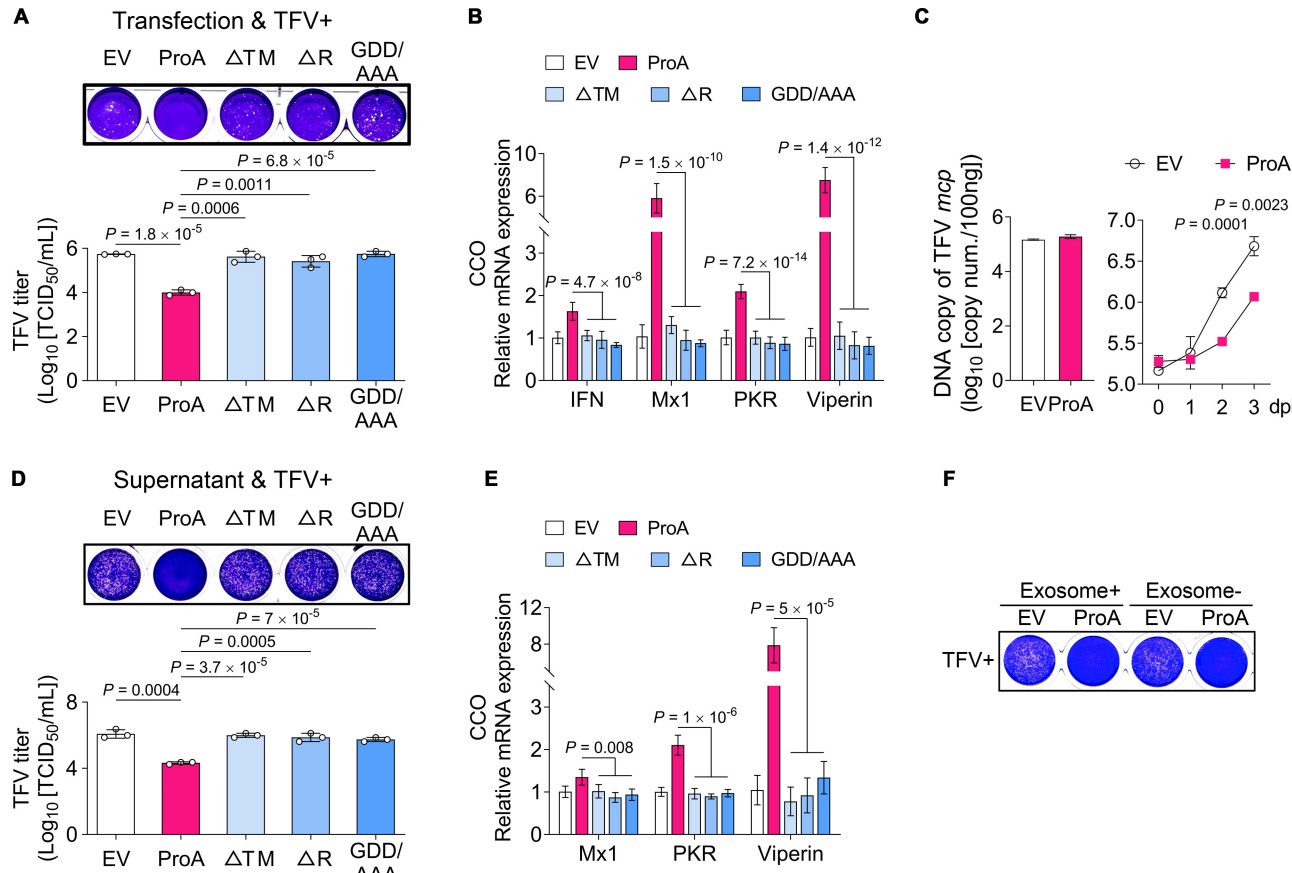

**FIG 2** ProA enhances the anti-TFV activity in CCO cells by upregulating IFN expression and secretion. (A) Overexpression of wild-type ProA suppressed TFV replication in CCO cells. Plasmids of ProA and indicated mutants were transfected into CCO cells 48 h before TFV infection [multiplicity of infection (MOI) = 2]. Crystal violet staining of infected monolayer cells and TFV titer detection were performed at 48 h post infection to evaluate the anti-TFV activity. (B) The mRNA levels of IFN and several ISGs in the samples mentioned in A were analyzed by real-time quantitative polymerase chain reaction (RT-qPCR). (C) Quantitative analysis of TFV *mcp* gene in different infection time points. The quantities of *mcp* were detected by qPCR. The left bar chart showed the copy number in EV- or ProA-transfected cell samples at 2 h post infection (hpi) which indicates the viral entry step. The right line chart represented the copy number variation during the viral replication step. (D) Supernatant from wild-type ProA-expression cells also suppressed TFV replication. The supernatants from plasmids of ProA and indicated mutants transfected CCO cells (24 h) were used to incubate with fresh CCO cells for 24 h, and TFV infection, crystal violet staining, and titer detection were performed as A mentioned. (E) The mRNA level of several ISGs in the samples mentioned in D was analyzed by RT-qPCR. (F) Exosome was not related to the anti-TFV activity. The supernatants from ProA- or EV-transfected CCO cells were used to collect (+) or remove (−) the exosomes; then, the products were incubated with fresh cells for 24 h. TFV infection and crystal violet staining were carried out 3 d post infection. For A to E, $n = 3$ independent experiments. Means ± SD are shown, and $P$ values were calculated using two-tailed unpaired Student's $t$-test.

their different IFN responses during ProA activation. As the RNA sensors in the RLR signal pathway, RIG-I and LGP2 were significantly upregulated by ProA overexpression in CCO (Fig. 3A), while all of the RLRs including MDA5 were upregulated by both ProA and IFN in FHM (Fig. 3B). Although the expression of MDA5 tended to be upregulated, its upregulation in COO was not significant. Accordingly, MDA5 of CCO was omitted in the following studies. In FHM, MDA5 expression was upregulated by ProA about five times compared with the EV sample, while expression of RIG-I and LGP2 was upregulated more than 60- and 15-fold by ProA, indicating that all of them are ISGs and that MDA5 is a less responsive RLR to ProA activation. Overexpression of RIG-I in CCO activated the IFN promoter about threefold, while RIG-I and ProA co-expression greatly increased the activation about 15-fold (Fig. 3C, left panel). The superimposed effect was stronger in FHM cells of RIG-I + ProA and MDA5 + ProA samples (Fig. 3C, right panel). Therefore, RIG-I and MDA5 can enhance the ProA-mediated IFN induction to form a positive feedback loop (Fig. 3D). LGP2 as the suppressing RLR could strongly inhibit IFN promoter when

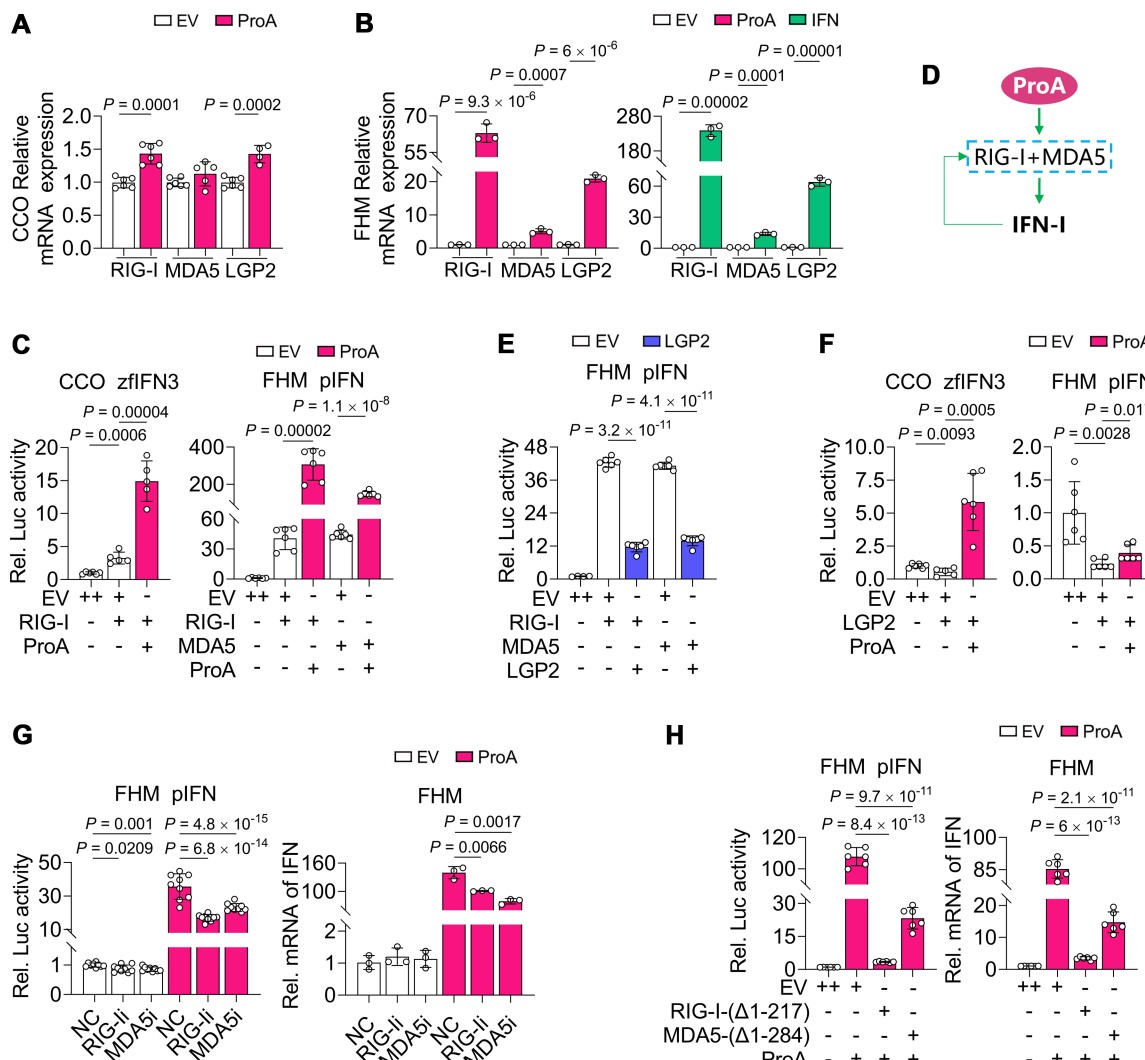

**FIG 3** The response and necessity of RIG-I, MDA5, and LGP2 to ProA overexpression in CCO and FHM cells. (A and B) The mRNA change of RLRs. Plasmids of ProA, IFN, or EV were transfected into CCO cells (A) and FHM cells (B), and the mRNA level of RIG-I, MDA5, and LGP2 was analyzed by RT-qPCR. (C) IFN promoter activity in cells of RIG-I or MDA5 expressing along and with ProA. Plasmids of ProA or EV were co-transfected with RIG-I plasmid in CCO cells and with RIG-I or MDA5 in FHM cells. Dual-luciferase reporter assay was performed to detect the IFN promoter activity. (D) A schematic diagram of ProA-mediated IFN expression and the positive feedback loop of IFN signaling in FHM cells. The blue box with the dashed line indicates the combination that the feature of RIG-I and MDA5 was the same. The green thick arrow and the green thin arrow represent the stimulatory relationship and the positive feedback loop. (E) Luciferase activities were determined from FHM cells co-transfected with LGP2 and RIG-I or MDA5 plasmids. (F) Luciferase activities were detected from CCO or FHM cells co-transfected with LGP2 and ProA plasmids. (G) Luciferase activities and IFN expression were evaluated in FHM cells co-transfected with siRNAs specific to RIG-I (RIG-Ii) or MDA5 (MDA5i) and plasmid of ProA. The siRNA targeting *gfp* sequence was utilized as the negative control (NC). (H) Luciferase activities and IFN expression were evaluated in FHM cells co-transfected with plasmids of ProA and EV, RIG-I-(Δ1-217), or MDA5-(Δ1-284) as indicated. For C, E, F, G, and H, luciferase reporter plasmids with different IFN promoters were used in corresponding cells as indicated. For C, E, F, and H, EV with double doses was used as negative control (++). All experiments were repeated at least three times independently, and all data are presented as means ± SD. *P* values were calculated using two-tailed unpaired Student's *t*-test.

co-expressed with RIG-I or MDA5 in FHM (Fig. 3E). In CCO cells, LGP2 could also suppress IFN promoter. However, this inhibition effect could be totally subverted by ProA with about a 10-fold increase of IFN promoter activity compared to the sample of LGP2 along and fivefold to EV control. In FHM, the IFN promoter activity of the co-expressed sample was still suppressed by LGP2, although ProA significantly triggered the activation (Fig. 3F). From the point of view of fish cells, LGP2 is a strong negative modulator in RLR

signaling. The response of these RNA sensors in FHM and CCO to ProA was identical, but the extent was different in these two cells and FHM was much stronger.

To identify the necessity of RLRs for ProA-mediated activation, siRNA knockdown of endogenous RIG-I and MDA5 in FHM could reduce the IFN promoter activity and IFN expression significantly (Fig. 3G). Unlike the mammalian cells, the efficiency of RNA interference in FHM was relatively low (Fig. S3). To further verify the importance of these two RLRs, the dominant negative mutants (DN) of RIG-I and MDA5, namely, RIG-I-(Δ1-217) and MDA5-(Δ1-284), were generated and transiently transfected into FHM before ProA activation. The results indicated that both IFN promoter activity and IFN expression were dramatically dropped (Fig. 3H). In all, RIG-I and MDA5 are important for ProA-mediated IFN expression and can promote this induction to form a positive feedback loop.

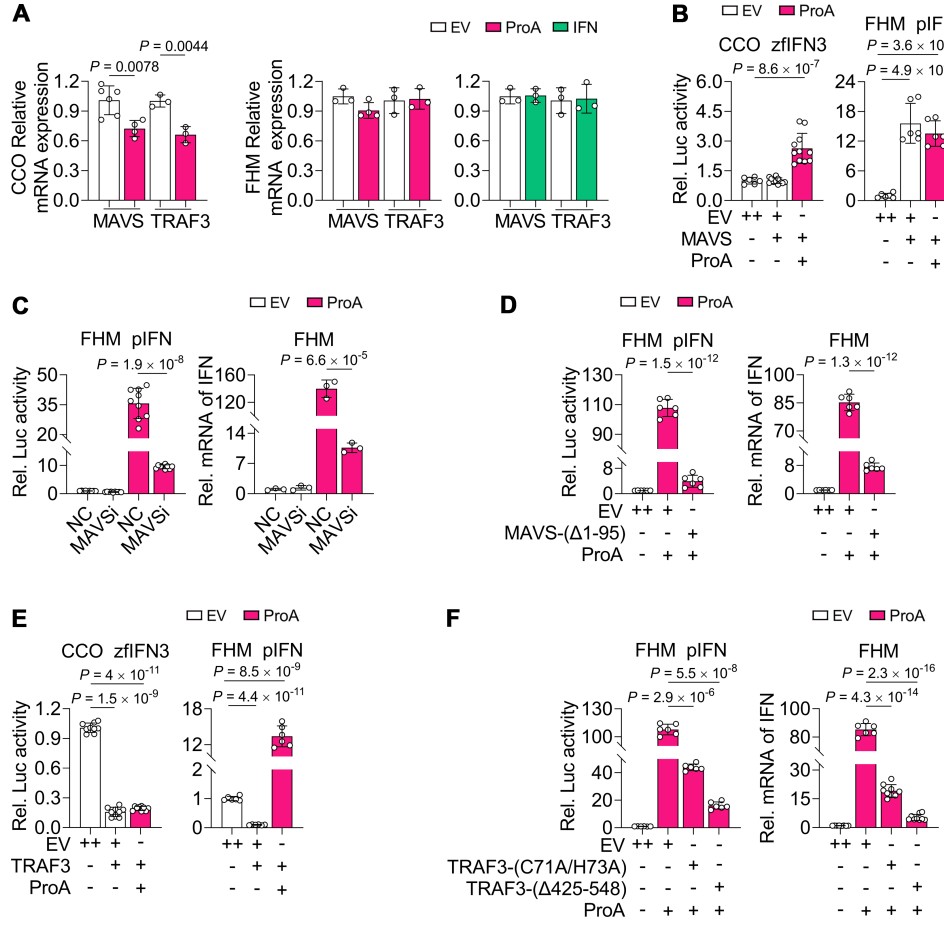

**FIG 4** MAVS and TRAF3 are required for ProA-mediated IFN production but not upregulated by ProA. (A) The mRNA level of MAVS and TRAF3 was not upregulated by ProA. The expression of MAVS and TRAF3 was detected by RT-qPCR in ProA plasmid-transfected CCO or FHM cells and IFN plasmid-transfected FHM cells. (B) IFN promoter activity in MAVS-transfected or MAVS and ProA co-transfected CCO or FHM cells. Luciferase activities were detected from CCO or FHM cells co-transfected with EV or ProA and MAVS plasmids. (C and D) Luciferase activities and IFN expression were evaluated in FHM cells co-transfected with siRNAs specific to MAVS and ProA plasmid (C) and co-transfected with plasmids of EV or MAVS-(Δ1-95) and ProA (D). (E) Luciferase activities were determined in CCO and FHM cells co-transfected with EV or ProA and TRAF3 plasmids. (F) Luciferase activities and IFN expression were evaluated in FHM cells co-transfected with plasmids of EV or TRAF3-(C71A/H73A), TRAF3-(Δ425-548), and ProA. For B, C, D, E, and F, luciferase reporter plasmids with different IFN promoters were used in corresponding cells as indicated. For B, D, E, and F, empty vector with double doses (++) was used as the negative control. All experiments were repeated at least three times independently, and all data are presented as means ± SD. P values were calculated using two-tailed unpaired Student's t-test.

## MAVS and TRAF are required for ProA-mediated IFN activation

Both MAVS, an adaptor for RIG-I/MDA5, and TRAF3, downstream of MAVS, could not be upregulated either by ProA in CCO and FHM or by IFN in FHM (Fig. 4A), suggesting that fish MAVS and TRAF3 are not ISGs. Overexpression of MAVS could not activate IFN promoter in CCO, but it could be promoted by co-expressing ProA. On the contrary, MAVS could activate the IFN promoter in FHM, but ProA could not enhance this activation (Fig. 4B). Moreover, MAVS was required for the ProA-mediated IFN promoter activation and IFN expression verified by both siRNA knockdown (Fig. 4C) and transfection of MAVS mutant, MAVS-(Δ1-95) (Fig. 4D). As for TRAF3, its overexpression could strongly suppress IFN promoter activity in both CCO and FHM. By ProA overexpression, this suppression could not be countered in CCO but could be reversed in FHM (Fig. 4E). The opposite IFN promoter reaction of MAVS and TRAF3 co-expressed with ProA in CCO and FHM cells suggested the different immune response intensity of RLR signaling in NNV-permissive and NNV-nonpermissive cells. In the DN overexpression samples of FHM, TRAF3-(C71A/H73A) holding amino acid substitution in the RING-finger domain exhibited less inhibition on both IFN promoter activity and IFN mRNA level when compared to TRAF3-(Δ425-548) with meprin and TRAF homology (MATH) domain deletion, suggesting that the RING-finger domain of TRAF3 is more important than the MATH domain in the signaling of ProA activation (Fig. 4F). In all, MAVS and TRAF3 are required for ProA-mediated IFN production but not upregulated by ProA.

## TBK1 phosphorylation and IRF3 nuclear translocation are necessary for ProA-mediated IFN activation

By overexpression of DN mutants, our previous study proved that TBK1 and IRF3 were essential for ProA-induced IFN promoter activation (30). Here, we showed that TBK1 and IRF3 could be upregulated by ProA in CCO and by ProA and IFN in FHM cells, indicating that they are ISGs. Notably, the extent of upregulation in CCO cells is significantly lower compared to FHM cells (Fig. 5A). In CCO cells, overexpressed TBK1 or IRF3 alone could not activate the IFN promoter, but they could induce luciferase activity when co-expressed with ProA. The same results could be observed in FHM cells except that TBK1 or IRF3 alone could activate the IFN promoter at a low level (Fig. 5B). To delineate the variation in phosphorylated and total protein content of TBK1 and IRF3 mediated by ProA, immunoblot analysis was conducted. First, the commercial antibodies were tested in fish cell samples, and the anti-phosphorylated TBK1 (pTBK1) antibody was verified to detect pTBK1 but not TBK1 of FHM cells (Fig. S4). Then, the results obtained from ProA-expressed FHM cells showed that about 2.9-fold upregulation of pTBK1 and 1.2-fold increase of IRF3 protein content in whole cell lysates were acquired when compared with the EV-transfected sample (Fig. 5C). Due to short of suitable antibodies detecting phosphorylated IRF3 of FHM cells, we performed nucleus and cytosol fractionation to determine the nuclear translocation of IRF3 which is another evidence for the activation of this transcription factor. The immunoblot analyses of the separated fractions proved that there was 1.3-fold augmentation in the cytoplasmic fraction and 1.8-fold increase in the nuclear fraction of ProA-expressed samples, indicating that the nuclear translocation of IRF3 was increased by ProA (Fig. 5D). The next step of IRF3 nuclear translocation, also the final step of RLR pathway, is to induce IFN expression, which was detected in ProA-expressed CCO cells (Fig. 5A). To sum up, ProA induces IFN expression via RIG-I/MDA5-MAVS-TRAF3-TBK1-IRF3 signal pathway in FHM cells.

## NOD1 and PKR are not needed but the negative regulatory factors for ProA-mediated IFN activation

Our previous research demonstrated that ProA could neither activate IFN promoter via the NF-κB response element nor upregulate the transcriptional levels of cytokine factors in FHM cells (30). Here, in CCO and FHM cells, the mRNA level of NF-κB was also not related to ProA overexpression (Fig. 6A). However, NOD1, a dsRNA sensor participating

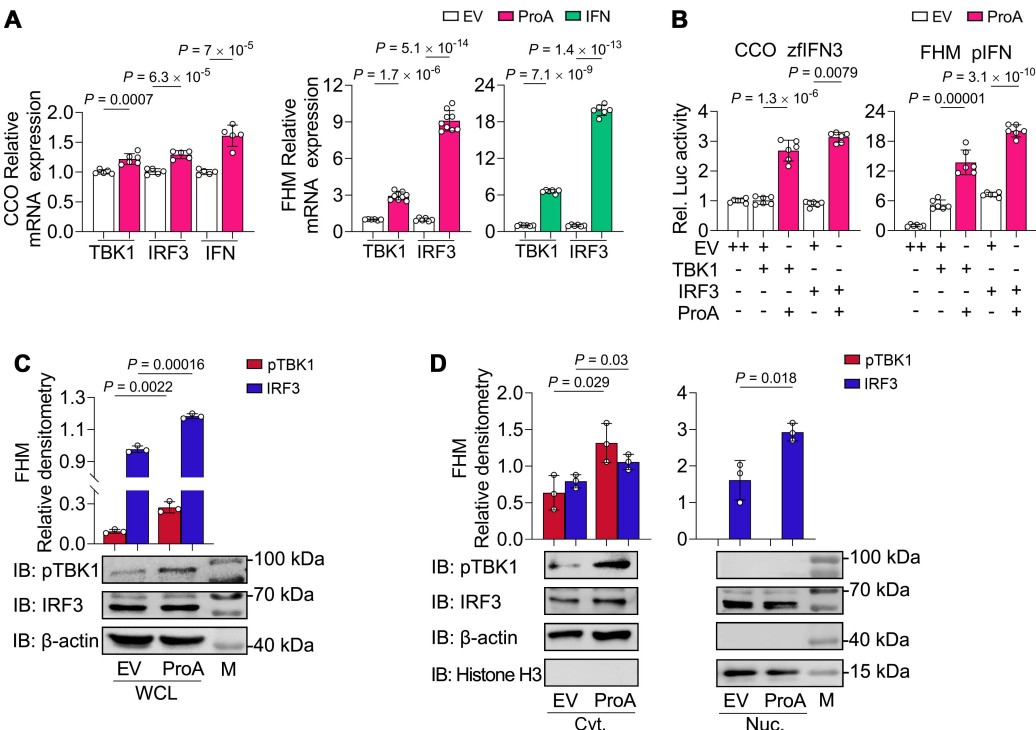

**FIG 5** ProA upregulates IFN expression via increasing TBK1 phosphorylation and IRF3 nuclear translocation. (A) The expression of TBK1, IRF3, and IFN in ProA- or IFN-expressing cells. Plasmids of EV, ProA, or IFN were transfected into CCO cells or FHM cells, and the mRNA level of TBK1, IRF3, and IFN was analyzed by RT-qPCR. (B) Variation of IFN promoter activity. IFN promoter activity was detected by dual-luciferase reporter assay in ProA or EV and TBK1 or IRF3 co-transfected CCO or FHM cells with different promoter reporters as indicated. EV with double doses (++) was used as the negative control. (C and D) Immunoblot analysis of protein level variation in FHM cells. The protein detection of pTBK1 and IRF3 in the whole cell lysate (WCL) (C) or cytoplasmic fraction (Cyt.) and nuclear fraction (Nuc.) (D) of FHM cells were conducted after plasmids of ProA or EV were transfected for 36 h. The intensity of validated bands was scanned, and the quantification of protein was normalized to those of the internal control protein of β-actin in the cytoplasm or histone H3 in the nucleus. For C and D, the bar charts showed the relative densitometry of bands of p-TBK1 and IRF3. The band size of the protein marker (M) was labeled on the right. All experiments were repeated at least three times independently, and all data are presented as means ± SD. *P* values were calculated using two-tailed unpaired Student's *t*-test.

in the innate immune response including IFN and inflammatory cytokines expression (13), was upregulated by ProA slightly in both CCO and FHM (Fig. 6A), indicating that NOD1 is also an ISG. In CCO, RIPK2, the adaptor for NOD1, activated IFN promoter activity only co-expressed with ProA for about 3.5-fold, but in contrast, NOD1 of FHM strikingly inhibited IFN activation, whether expressed alone or co-expressed with ProA (Fig. 6B). The inhibition led us to evaluate its suppression on the activating RLRs, RIG-I and MDA5, when co-expressed in FHM. The results showed that NOD1 strongly inhibited IFN promoter activity induced by RIG-I or MDA5, and the luciferase activities of the NOD1-MDA5 samples were about 63% of that of LGP2-MDA5 samples (Fig. 6C). By siRNA knockdown of NOD1, no attenuation of the IFN expression was observed, suggesting that NOD1 was not necessary for ProA-mediated IFN activation (Fig. 6D). Moreover, another RNA sensor, PKR, has already proved to be an ISG (30). It also inhibited IFN promoter activity with stronger suppression than NOD1 (Fig. 6E). ProA-mediated luciferase activities were reduced to 20% by PKR and 41% by NOD1. Therefore, NOD1 and PKR are not needed for ProA-mediated IFN activation, whereas they are negative modulators of the IFN signaling at the RNA sensor level.

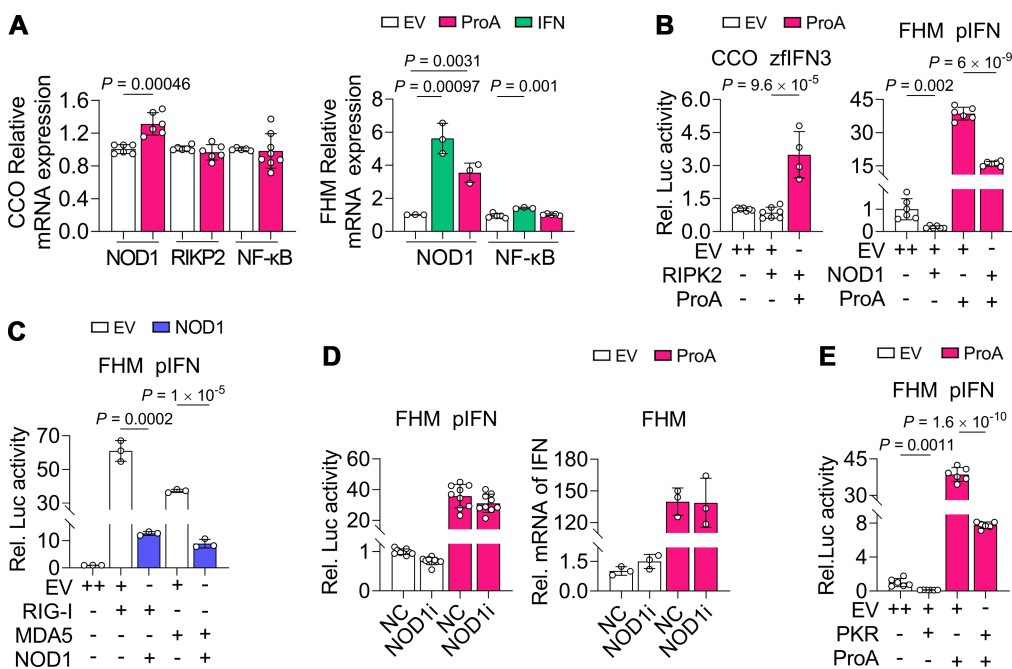

**FIG 6** NOD1 and PKR are negative regulatory factors for ProA-induced IFN expression. (A) The transcription of NOD1, RIPK2, and NF-κB in ProA- or IFN-expressing cells. The mRNA level of these proteins was detected by RT-qPCR in EV, ProA, or IFN plasmids transfected CCO or FHM cells. (B) Changes in IFN promoter activity. Plasmids of ProA or EV were co-transfected with RIPK2 plasmid in CCO cells or with NOD1 in FHM cells. Dual-luciferase reporter assay was performed to detect the IFN promoter activity with different promoter reporters as indicated. (C) Luciferase activities were determined in FHM cells co-expressing with NOD1 and RIG-I or MDA5. (D) Luciferase activities and IFN expression were evaluated in FHM cells co-transfected with siRNAs specific to NOD1 (NOD1i) or *gfp* sequence (NC) and plasmid of EV or ProA. (E) Luciferase activities were detected from FHM cells co-expressing with ProA or EV and PKR. For B, C, and E, EV with double doses (++) was used as the negative control. All experiments were repeated at least three times independently, and all data are presented as means ± SD. *P* values were calculated using two-tailed unpaired Student's *t*-test.

## TLR3 and TRIF are irrelevant to ProA-mediated IFN activation

TLR3 is a dsRNA sensor that is important for IFN-β expression induced by hepatitis C virus NS5B (31). Here, in CCO cells, TLR3 and its adaptor, TRIF, were both upregulated slightly by ProA (Fig. 7A). ProA only upregulated TLR3 in FHM cells, while IFN could upregulate the expression of both TLR3 and TRIF (Fig. 7B), indicating that they are ISGs. To investigate whether the TLR3 signaling brunch is necessary for ProA-mediated IFN expression, ProA and TRIF-(302-459) only with the TIR domain were co-expressed in FHM, and then, TRIF-(302-459) could not interfere with the IFN upregulation mediated by ProA. Consequently, the signal transduction of ProA-mediated IFN response does not go through the TLR-TRIF-NF-κB branch.

## NNV infection suppresses the ProA-mediated IFN activation in CCO cells

The above results demonstrated that the intensity of immune response triggered by ProA in FHM is much stronger than that in CCO, and RIG-I and MDA5 might be the key regulators. To figure out the mRNA expression of the mentioned factors of these signaling pathways in the state of NNV replication, CCO cells were transfected with the plasmid of ProA followed by NNV infection. From the RT-qPCR results, neither all the factors in ProA-related RLR signaling (Fig. 8A through C), including RLRs (RIG-I, MDA5, and LGP2), adaptors (MAVS, TRAF3, and TBK1), transcription factor (IRF3), and IFN nor the ProA-unrelated factors, including NOD1 and TLR3 brunches (Fig. 8D), could be upregulated by ProA. The expression of positive modulators, RIG-I and MDA5, was reduced, while LGP2 was maintained at the same level at 48 h post infection when ProA-expressing

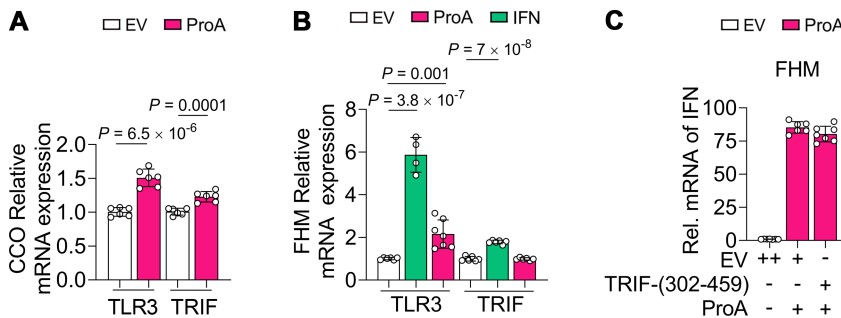

**FIG 7** TLR3 and TRIF are not irrelevant to ProA-mediated IFN activation. (A and B) RT-PCR detection for transcription of TLR3 and TRIF in EV-, ProA-, or IFN-expressed cells. Plasmids of EV, ProA, and IFN were transfected into CCO cells (A) and FHM cells (B) to detect the expression of TLR3 and TRIF by RT-qPCR. (C) The mRNA level of IFN was determined in FHM cells co-expressed with EV or TRIF-(302-459) and ProA. EV with double doses (++) was used as the negative control. All experiments were repeated at least three times independently, and all data are presented as means ± SD. $P$ values were calculated using two-tailed unpaired Student's $t$-test.

CCO cells were infected (Fig. 8A), implying that there is another mechanism from NNV to suppress the immune stimulation during replication. By comparison, FHM cells were also performed NNV infection after wild-type or DN mutants of RIG-I and MDA5 transfection to validate the antiviral role of RIG-I and MDA5 for NNV proliferation. The viral quantities represented by CP content in DN-transfected cells were about twofold higher than that of EV control cells with the most amount in the RIG-I-($\Delta$1-217) sample (24 h), suggesting that the blockage of RIG-I and MDA5 facilitated NNV infection. The antiviral role of RIG-I and MDA5 can be further proven that CP expression in RIG-I- and MDA5-overexpressed cells were much lower than that in EV-transfected cells (Fig. 8E). However, productive NNV infection could not be achieved in DN-transfected FHM cells because CP quantity could not increase with prolonged infection time (48 and 72 h), which is a typical feature of unsuccessful viral replication. Consequently, RIG-I is the key node in RLR signaling for ProA activation and NNV suppression.

Although the components were slightly different in FHM (Fig. 8F) and CCO cells (Fig. 8G), the required signal pathways and negative regulators of ProA-mediated IFN activation were almost identical in these fish cells. Moreover, the intensity of immune response in these two cells is different, which could be figured out by the digits under the factor names in the schematic diagrams. The expression fold change of all factors upregulated by ProA in FHM was significantly larger than that of CCO (the red digits of Fig. 8F and G), and therein, IFN expression in FHM was about 24-fold stronger than that in CCO. On the other hand, the upregulation of FHM RIG-I response to both ProA and IFN was at least threefold higher than that of the negative regulators of LGP2, NOD1, and PKR, implying that ProA-mediated IFN activation is stronger than the suppression from these dsRNA sensors. Based on the facts of IFN signaling intensity and the sensitivity to NNV, we meditated that the IFN response intensity may determine the productive infection of NNV.

## DISCUSSION

NNV, causing huge continual economic losses in the global fish fry industry, is an important aquatic virus whose research is also an area that cannot be ignored. Studies on the innate immunity of fish not only offer practical targets for preventing and controlling fish diseases but also provide fundamental information for comparing the evolutionary aspects of lower vertebrates, fish, and higher vertebrates, such as mammals. The application of advanced technology in fish disease research has led to the accumulation of a substantial amount of data regarding the immune response of fish to fish viruses (32).

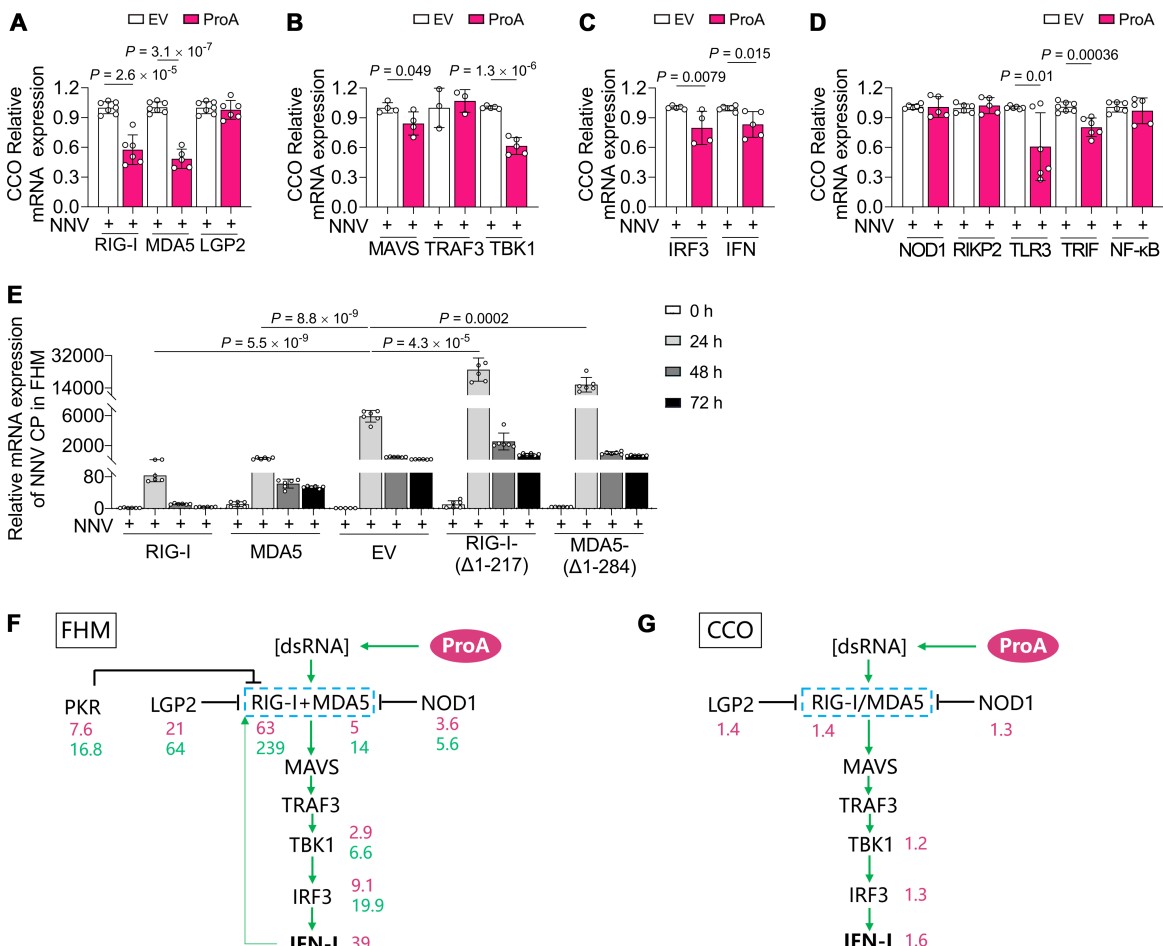

FIG 8  NNV infection suppresses the ProA-mediated IFN activation. (A–D) The mRNA variation of tested factors in CCO cells infected with NNV. Plasmids of ProA or EV were transfected into CCO cells followed by NNV infection (MOI = 10). The mRNA level of RLRs (A), adaptors (B), transcription factor and IFN (C), and the ProA-unrelated factors (D) was analyzed by RT-qPCR at 48 hpi. (E) NNV replication level in transfected FHM cells. CP expression representing NNV replication was evaluated by RT-qPCR in FHM cells transfected with RIG-I, MDA5, EV, RIG-I-(Δ1-217), and MDA5-(Δ1-284) at different infection time points as indicated. (F and G) The schematic diagrams of ProA-triggered IFN-related signal pathways in FHM (F) and CCO (G) cells. The blue box with the dashed line indicates the combination that the feature of RIG-I and MDA5 was the same. The green thick arrows and the black inhibition lines indicate stimulatory and inhibitory relationships. The green thin arrow represents the positive feedback loop. The red and green digits are the expression fold changes upregulated by ProA and IFN, respectively. For A, B, C, D, and E, NNV infection was performed at MOI = 10 after 24 h post plasmid transfection. All experiments were repeated at least three times independently, and all data are presented as means ± SD. $P$ values were calculated using two-tailed unpaired Student's $t$-test.

The innate immune system detects invading viruses mainly by recognizing the pathogenic nucleic acid through PRRs. RLRs, including RIG-I and MDA5, have displayed the importance of viral RNA sensing (33). Multiple viral proteins have been discovered to modulate the host's innate immune system, including the regulation of IFN signaling, which can be activated or suppressed to facilitate immune evasion, productive replication, and successful release. RLR-related IFN signaling was reported as the main responsive pathway of host immunity for NNV (34, 35) and other aquatic RNA viruses (36, 37) and also the key target for being regulated by viruses to achieve immune evasion (38). RdRp, the essential protein of RNA viruses, is responsible for replicating their RNA genomes. When it is overexpressed in the host cell but not in the state of viral infection, RdRp also exerts RNA replication activity to generate double-stranded intermediates that trigger an RLR-related IFN response (28, 29). Here, our experimental results show that ProA, the RdRp encoded by OGNNV, induced IFN response by its RNA replication activity. Furthermore, ProA-triggered IFN promoter activation may be specific for fish cells. Silencing essential components of the RLR signaling pathway using siRNA

or negative mutants confirmed the dependence of ProA-induced IFN activation on the RIG-I/MDA5-MAVS-TRAF3-TBK1-IRF3 pathway. Thereinto, RIG-I is the key node for this immune response as identified by the upregulation level in ProA-expressed FHM (Fig. 8F) and the enhancement of NNV replication in RIG-I-(Δ1-217)-expressed NNV-nonpermissive FHM cells (Fig. 8E). Also, RIG-I is the key target for NNV suppression as determined by the expression inhibition in ProA-expressed and NNV-infected CCO cells (Fig. 8A).

The RLR signaling is mainly regulated through the RNA sensors level. During ProA-mediated IFN activation, the responses of RIG-I including the mRNA expression level to ProA or IFN and the synergism with ProA to activate IFN promoter activity were stronger than that of MDA5 in both FHM and CCO cells, suggesting that MDA5 may be low sensitivity to the ProA-generated dsRNA which is mainly recognized by RIG-I. These differences may be due to the recognition of the short, blunt end, and 5′-triphosphate group (5′ppp) dsRNA by RIG-I and the binding of long RNA with secondary structure by MDA5 (4). Lots of research on RIG-I agonists indicated that RIG-I plays a key role in the recognition of RNA viruses including Sendai virus (SeV) (39, 40), reovirus (41), vesicular stomatitis virus (42), Newcastle disease virus (43), and influenza virus (44). Relatively fewer reports investigated the activation of MDA5 by viruses such as encephalomyocarditis virus (45, 46), a picornavirus. Infection of some RNA viruses, such as dengue virus and SeV, can activate both RIG-I and MDA5 (47, 48). For most RNA viruses, MDA5 seems to be functionally redundant (49). Recently, MDA5 was found to detect cellular RNAs with a distinctive feature, suggesting that MDA5 should be studied in the contexts of different virus infections, autoinflammation, and cancer (50). Besides RIG-I and MDA5, other RLR such as LGP2 and other PRRs like NOD1, PKR, and TLR3 are all dsRNA sensors and may take effect to activate IFN response (4). It has been proven that LGP2 overexpression negatively regulates the antiviral response in fish cells (51). We obtained the same results in this study and demonstrated that the negative regulation of RIG-I and MDA5 was mediated at the sensor level. In addition to the well-characterized role in sensing bacteria, mammalian NOD1 also takes part in sensing RNA viruses (52). In zebrafish, NOD1 not only plays cumulative effects with MDA5 to enhance the antiviral response of RLR-mediated signaling but also antagonizes the truncated isoform of MDA5 (53). However, we found that NOD1 in FHM cells inhibited ProA-, RIG-I-, and MDA5-mediated IFN promoter activation. Moreover, NOD1 exhibited stronger suppression to MDA5 than to RIG-I. We speculate that the recognition RNA feature of NOD1 and MDA5 may be similar and their interaction deserves further study. Furthermore, the PKR-expressed sample exhibited the most pronounced suppression of ProA-mediated activation of the IFN promoter suggesting that PKR may play as a negative regulator for RLR signaling. This suppression may be due to the preferential binding of PKR to circular RNA (54). As for TLR3, no effect was observed in the TRIF-(302-459)-transfected cells. In line with this result, TLR3 mainly expresses and functions on professional phagocytes (55) as an IRF3- and IFN-inducible factor (56). Therefore, both NOD1 and TLR3 branches are not related to ProA-mediated IFN activation. Our results enrich the members of PRR in teleost (57) as well as the regulation information for RLR signaling (38).

The intensity of immune response to dsRNA in different fish cells may be related to the robust proliferation of NNV. The lower the response intensity, the more vigorous replication of the NNV. This is also in accord with the distribution of active NNV replication in fish (58) where the brain and eyes, the nervous system with low innate immunity (59), are the main targets for infection (21). From the IFN promoter assay, the strength of ProA-mediated luciferase activity in FHM is much higher than that in CCO cells. By RNAi experiment, the replication of NNV was enhanced by knocking down the main antiviral PRR, RIG-I. Furthermore, the transcriptome data revealed that the immune response of permissive cells, SSN-1 (60) and CCO (unpublished data), during NNV infection was weak, as the expression fold change of the immune-related genes in the main pathways including upregulation and downregulation was not larger than fivefold.

In general, a cell line that supports productive viral infection must be susceptible and permissive to the virus. For susceptible cells, they contain the receptors that allow viral attachment and entry, exhibiting the ability to support the absorption of viruses (61, 62). Both FHM and CCO cells in this study are susceptible to NNV. For permissive cells, they allow the completion of the viral cycle and the release of viral progeny, possibly as they have several abnormalities in their antiviral intrinsic immunity factors (63) or restriction factors for viral traffic (64). For instance, the CCO cells that are weak in IFN response could well support NNV replication while FHM only supported a low level of NNV replication when RIG-I and MDA5 were partially silencing. Therefore, we meditate that the level of IFN response to ProA is one of the determinants for the successful replication of NNV.

Besides the weak IFN response from the permissive cells, there are other IFN inhibition methods from NNV. First, NNV infection can upregulate the negative regulators from the host, such as Nectin4 (65) and BAG3 (66), to achieve IFN suppression. Nectin4 interacts with CP and TRAF3 to promote TRAF3 degradation by autophagy-mediated lysosomal pathway to inhibit the RLR pathway. BAG3 upregulates LC3-II and downregulates RLR and NF-κB signal pathways to promote NNV replication. In this study, we found that ProA-mediated factors (RIG-I, MDA5, TBK1, IRF3, and IFN) were downregulated in CCO cells during authentic NNV infection, while the negative regulators (LGP2 and TRAF3) maintained their expression. Second, NNV encodes B1, B2, and CP to obtain a potent inhibitory effect on the immune response. By dysregulating RNA polymerase II, B1 (67) and B2 (34) block the host's IFN response. CP recruits E3 ubiquitin protease to reduce TRAF3 and interact to degrade IRF3 via ubiquitination (27). Therefore, it is important to investigate the virus-caused inhibitory mechanism targeting the host immune response.

Based on our previous results (30) and the current data, we put forward a hypothesis that ProA produces dsRNA during NNV replication, and the synthesized intermediates are mainly detected by RIG-I-like receptors (RIG-I and MDA5), resulting in IFN transcription through the essential signaling of RIG-I/MDA5-MAVS-TRAF3-TBK1-IRF3. To suppress the IFN response, negative regulation must be taken by NNV via modulating host factors or expressing viral protein. We identified that LGP2, NOD1, and PKR play a negative regulatory role in ProA-mediated IFN transcription at the RNA sensor level. This work revealed the mechanism of the interaction between NNV proteins and host innate immune signaling, providing fundamental information on target candidates for the prevention and control of aquatic viruses.

## MATERIALS AND METHODS

### Cell culture and viruses

Channel catfish (*Ictalurus punetaus*) ovary cells and striped snakehead cell (SSN-1) sub-clone no. 1 (SC1) cells were maintained in L-15 medium, FHM cells were cultured in M199 medium (Gibco), Asian SB cells were maintained in minimal essential medium, and zebrafish embryonic fibroblast (ZF4) cells (ATCC #CRL-2050) were cultured in Dulbecco's modified Eagle medium (DMEM)/F12 medium. All these fish cell lines are maintained in indicated medium supplemented with 10% (vol/vol) fetal bovine serum (FBS) at 27°C. HEK 293T, HeLa, and HepG2 cells were grown in DMEM supplemented with 10% FBS (Hyclone), penicillin (100 U/mL), and streptomycin (100 μg/mL) at 37°C with 5% $CO_2$. OGNNV isolated from moribund orange-spotted grouper larvae collected in the Hainan province of China was propagated and titrated in SC1 cells. TFV isolated from diseased tiger frog (*Rana tigrina rugulosa*) tadpoles were propagated and titrated in FHM cells. Both viruses were preserved in our laboratory.

### Virus infection, titer determination, and crystal violet staining

For NNV infection, non-transfected or transfected CCO cells were infected with OGNNV [multiplicity of infection (MOI) = 10] for 1 h, then incubated with fresh medium with 5% FBS, and collected at 48 h post infection for RT-qPCR. For TFV entry assay, transfected CCO cells were incubated with TFV (MOI = 2) for 2 h, washed by 1× phosphate buffer

solution (PBS) for three times, and collected for qPCR. TFV infection (MOI = 2), crystal violet staining of CCO cells, the anti-TFV assay of culture supernatant, and the titer assay of TFV were performed as previously described (30). $TCID_{50}$ (50% tissue culture infective dose) was calculated by the Reed-Muench method (68).

## Plasmid construction

The expression plasmids of pN3-Flag (EV control), ProA, and FHM genes including RIG-I, TBK1, and IRF3 were constructed in the previous study (30). After amplification from FHM cDNA, the open reading frame(ORF) of MDA5 was cloned into pCMV-N-Flag, and ORFs of IFN, LGP2, MAVS, TRAF3, NOD1, and PKR were inserted into pN3-Flag (30). The DN mutant of RIG-I (Δ1-217), MDA5 (Δ1-284), and MAVS (Δ1-95) were generated by replacing the original N-terminal CARD domains with the Flag tag. TRAF3 (C71A/H73A) was generated by substituting cysteine and histidine with alanine within the RING-finger domain, while TRAF3 (Δ425-548) was obtained by deleting the MATH domain. TRIF-(302-459) was constructed by inserting the fragment of TIR domain of FHM into pMCV-N-Flag. All the numbers in DN's names indicate the amino acid positions. All CCO genes including IFN, RIG-I, MDA5, LGP2, MAVS, TRAF3, TBK1, IRF3, and RIPK2 were amplified from CCO cDNA and inserted into pN3-Flag to generate N-/C-terminal Flag-tagged recombinant vectors as indicated in Table 2. The reporter plasmids, zfIFN1 and zfIFN3, were constructed by inserting the zebrafish IFN1 promoter sequence and zebrafish IFN3 promoter sequence into the pGL3-basic vector (69).

## Dual-luciferase reporter assay

FHM cells were trypsinized, adjusted density to $1 \times 10^5$, transiently transfected with various expression plasmids or EV (1 µg) together with pIFN (100 ng) and pRLcmv (20 ng) using InstantFECT Transfection Reagent (PGR-Solution) following the manufacturer's protocol, and seeded into 24-well plates. For CCO, SB, and ZF4 cells, all conditions were the same as mentioned above except that pIFN was replaced by zfIFN1 or zfIFN3. Mammalian cells were transfected with 50-ng pISRE-TA-luc and 10-ng pRLTK plus 1-µg ProA, EV, or hMAVS expression vector using Fugene Transfection Reagent (Promega). At the indicated time (24 h or 48 h) post transfection (hpt), cells were harvested, and the Firefly and Renilla luciferase activities were determined with the dual-luciferase reporter assay system (Promega) according to the manufacturer's instructions. Data are normalized for transfection efficiency by dividing Firefly luciferase activity by Renilla luciferase activity. Each experiment was repeated at least three times.

## Optical and fluorescent microscopy and cytotoxicity analysis

For determination of NNV or TFV infection in CCO cells, cytopathic effects of infected or transfected plus infected cells were observed using Leica TCS SP5 confocal laser microscope under optical microscopy. 293T cells were transfected with the expression plasmid for Flag-tagged ProA. After 48 h, cells were fixed in 4% paraformaldehyde and permeabilized with 0.1% Triton-X 100, then blocked for 30 min with 10% goat serum, and incubated overnight at 4°C with anti-Flag mouse monoclonal antibody (1:200; ab18230, Abcam). On the second day, the expression and localization of ProA-Flag were determined using Alexa fluor 488 donkey anti-mouse IgG (1:2,000; A-21202, Thermo Fisher) secondary antibody under fluorescent microscopy. CCO cells grown in 96-well plates were transfected with expression plasmids of EV or ProA at 100 ng or 1,000 ng for 48 h. Then, cell viability was assayed using Cell Counting Kit-8 (CCK8, Beyotime) according to the manufacturer's instructions.

## RNA isolation, reverse transcriptase PCR (RT-PCR), and real-time quantitative PCR (RT-qPCR)

The total RNA was purified using the RNeasy mini kit (Qiagen, Germany) according to recommended protocol and treated with gDNA Eraser. The first strand of cDNA was

**TABLE 2** Primers used in this study

| Name | Sequence (5′–3′) | Remark[a] |
|---|---|---|
| Cloning primers | | |
| CCO-IFN-F | CCGCTCGAGCCGCCACCATGGACATCAAACTGTCATG | MG799351 |
| CCO-IFN-R | CCCAAGCTTGGTTGGTCCTTTCCAGTAGTTTC | |
| CCO-RIG-I-F | TCCCCGCGGACCATGGATTACAAGGATGACGACGATAAGGGATCCATCGCCTACGAGTCGGAG | N-terminal Flag tag added; OP805912 |
| CCO-RIG-I-R | CAGAGACGACGTGGGAGCCAAGAACCTAGGCGGCCGCAT | |
| CCO-MDA5-F | TCCCCGCGGACCATGGACCATGAGCAGGAGAAGAAAACCATCG | OP805911 |
| CCO-MDA5-R | GATGAGATGGACGGATCCATCGCCGATTACAAGGATGACGACGATAAGTAAGCGGCCGCAT | C-terminal Flag tag added |
| CCO-LGP2-F | CCCAAGCTTACCATGGAGAAGATATCACTGAGG | OP805909 |
| CCO-LGP2-R | GGAATTCGAGTCCTGCTCGAGGTCA | |
| CCO-MAVS-F | GGAATTCACCATGGATTACAAGGATGACGACGATAAGGGATCCATCGCCGCATATGCAGGT | N-terminal Flag tag added; OP805910 |
| CCO-MAVS-R | ATGCGGCCGCTTAGTACTTGAGCCTCCAGGCCACAAAAACAG | |
| CCO-TRAF3-F | GCAGTCGACGCCACCATGTCCTTGCCGCGCC | OP805914 |
| CCO-TRAF3-R | CCGGGCCCCCTGGGTCAGGCAGGTCTGAG | |
| CCO-TBK1-F | GGAATTCACCATGGTGCAGAGTACGGCGAACTACCTGTG | OP846095 |
| CCO-TBK1-R | ATGCGGCCGCTTACTTATCGTCGTCATCCTTGTAATCGGCGATGGATCCGATTCTGTCCAC | |
| CCO-IRF3-F | TCCCCGCGGACCATGGCTCAGCCCAAACCACTCTTCATTC | OP846094 |
| CCO-IRF3-R | ATGCGGCCGCTTACTTATCGTCGTCATCCTTGTAATCGGCGATGGATCCCCACAGTTCCAT | |
| CCO-RIPK2-F | CCGCTCGAGACCATGGACCAGGCGGGCT | OP805913 |
| CCO-RIPK2-R | GGGGTACCGAAGATGTTGCGGATGTTATTGC | |
| FHM-IFN-L | CCCAAGCTTGCCACCATGAAAACTCAAATGTGGACGTATAT | FN178457 |
| FHM-IFN-R | CGGGGTACCTCGTCTGTTGGCAATGCTTG | |
| FHM-LGP2-F | CCCAAGCTTACCATGGAGATCGCTCTTAGA | OP805915 |
| FHM-LGP2-R | GGAATTCGATAAGTCCAGGTCAGGGTAA | |
| FHM-MAVS-F | GAAGCTTCCGCCACCATGGATTACAAGGATGACGACGATAAGATGTCACTGAC | N-terminal Flag tag added; FN178455 |
| FHM-MAVS-R | ATAAGAATGCGGCCGCTTAATGATTGAGCTTCCAGGCCAAGAAAACAGC | |
| FHM-TRAF3-F | CCCAAGCTTGCCACCATGTCCGCAGGGCGTAATGT | XM_039669301 |
| FHM-TRAF3-R | GAAGTCGACAGGGTCCGGGAGGTCAGAG | |
| FHM RIG-I-(Δ1-217)-F | GGAATTCCCGCCACCATGGATTACAAGGATGACGACGATAAGACGTTCAGAGAGGAG | N-terminal Flag tag added & CARD domain (1–217 aa) deleted |
| FHM RIG-I-(Δ1-217)-R | ATAAGAATGCGGCCGCTCAGTCTCTCAGCGGCCATGTTTGAGGC | FN394062 |
| FHM-MDA5-(Δ1-284)-F | CCCAAGCTTACCATGTCAGCGAGGACAAT | CARD domain (1–284 aa) deleted |
| FHM-MDA5-(Δ1-284)-R | GGGGTACCGTCGCCGTCCGTGTC | MG799354 |
| FHM-MAVS-(Δ1-95)-F | CCCAAGCTTACCATGACGATCAGAGGAATC | CARD domain (1–95 aa) deleted |
| FHM-MAVS-(Δ1-95)-R | GGGGTACCATGATTGAGCTTCCAGGC | |
| FHM-TRAF3-(C71A/H73A)-F | GCTCTGTATACCTCGGCAAACTGAGGCCGGAGCACGCTTCTGCGAGAGC | Point mutations of C71A and H73A |
| FHM-TRAF3-(C71A/H73A)-R | CTGTGATGCAGCTCTCGCAGAAGCGTGCTCCGGCCTCAGTTTGCCGAGG | |
| FHM-TRAF3-(Δ425-548)-F | GCTAGAGACGGCCAGCTTCAATGGTACCCTCATTGAGAACGGAACCTACATCAAAGACG | MATH domain (425–548 aa) deleted |
| FHM-TRAF3-(Δ425-548)-R | GAAGATGGTATCGTCTTTGATGTAGGTTCCGTTCTCAATGAGGGTACCATTGAAGCTGG | |
| FHM-TRIF-(302-459)-F | CCCAAGCTTAACAGTTTTTGCTCTCCAGAAG | Part of TIR domain (302–459 aa) cloned |
| FHM-TRIF-(302-459)-R | GCTCTAGATCACCTTTGCTTTTCCACGTTGTCCG | MG799355 |
| FHM-TBK1-L | CCCAAGCTTGCCACCATGCAGAGTACGGCGAACTACC | LT174673 |
| FHM-TBK1-R | CGGGGTACCGATCCGGTCCACGTTCCTG | |
| FHM-IRF3-L | CCCAAGCTTGCCACCATGACTCAAGCAAAACCGCTG | HE856621 |

(*Continued on next page*)

**TABLE 2** Primers used in this study (*Continued*)

| Name | Sequence (5′–3′) | Remark[a] |
|---|---|---|
| FHM-IRF3-R | CGGGGTACCGCAGAGCTCCATCATTTGCTC | |
| FHM-NOD1-L | CCGGAATTCGCCACCATGGGGTCTTTCAAGAAGGTGT | MG799353 |
| FHM-NOD1-R | CGGGGTACCGCGGAAGCGAAGCCTCTT | |
| FHM-PKR-L | CCCAAGCTTGCCACCATGGAGTCTCCGTCAAGAAATTATAT | MG799352 |
| FHM-PKR-R | CGGGGTACCTGAGCTTGTGCTCAAAGCTAAAT | |
| qPCR primers | | |
| CCO-RIG-I-QF | CCCTCGCTGATGCTCTGAAA | |
| CCO-RIG-I-QR | ACCTGAGCCTGTCAGTTGTG | |
| CCO-MDA5-QF | AGATCCTCGGACTCACTGCT | |
| CCO-MDA5-QR | AGGCGTCCAAGTTAGCACAA | |
| CCO-LGP2-QF | CCGTGGAGACGACATGAGAC | |
| CCO-LGP2-QR | GCAACGCAGCTAATGGTACG | |
| CCO-MAVS-QF | CTTACTTGCCCTGCCTCACA | |
| CCO-MAVS-QR | TCTGTTCGCATGTCCGAAGT | |
| CCO-TRAF3-QF | ACAGCAGGTTACGGACCATT | |
| CCO-TRAF3-QR | TGGCCTCACGGTATTTGCAT | |
| CCO-TBK1-QF | TCATGCGTGTGATTGGGGAA | |
| CCO-TBK1-QR | CTCCTCCGTCCCATACAACG | |
| CCO-IRF3-QF | CCCGAGGCTTCAAGATGGTT | |
| CCO-IRF3-QR | ATGCCCCAGAATGTGAGTCG | |
| CCO-IFN-QF | AAATGGGACAGCAGGACACT | |
| CCO-IFN-QR | GACGCTGTACCACACTCCTG | |
| CCO-NOD1-QF | GGCGAGGTGTTCAGCTACAT | |
| CCO-NOD1-QR | CAGGAGACACCACTTCTGGC | |
| CCO-RIPK2-QF | ACCGCTAATCCTGACGAACG | |
| CCO-RIPK2-QR | CGATTTCGTCAAACCTCCGC | |
| CCO-TLR3-QF | GTCCAACCTGACCGAGCTTT | |
| CCO-TLR3-QR | GCAGACGTAGCTTTTGGCAC | |
| CCO-TRIF-QF | TGCATGCAGAGGAAGACTCA | |
| CCO-TRIF-QR | CGAGTTCTCAATGGCGTCCT | |
| CCO-NF-κB-QF | CGAACACGACAACATCTCGC | |
| CCO-NF-κB-QR | GCGATGTGAAGAGGTGTGGA | |
| FHM-RIG-I-QF | AACATCGAGCATCTGGCGAA | |
| FHM-RIG-I-QR | CTGCAGCTCTTCTGAACCGA | |
| FHM-MDA5-QF | CGCAGGGAATCTTATGGGCA | |
| FHM-MDA5-QR | GGGGCTCGATGATGCTGTAT | |
| FHM-LGP2-QF | TTGTCCCTCTCAGGGATCAGG | |
| FHM-LGP2-QR | GGGTCATAAGGAGCTTGGACTC | |
| FHM-MAVS-QF | GCCAGAGGAAGACCACTACG | |
| FHM-MAVS-QR | TGTGCCACTATGGTTCAGGG | |
| FHM-TRAF3-QF | CCACACTAGAGTCCAAGGTCG | |
| FHM-TRAF3-QR | GCTCCAGCTGCCTGTACTTT | |
| FHM-TBK1-QF | CTTCAGAGTCTCCTCACGCC | |
| FHM-TBK1-QR | ACTGGTCGAATCCCCAACAC | |
| FHM-IRF3-QF | CCCCAAAATGAAACCGTGGG | |
| FHM-IRF3-QR | ACGTGTTCAAACCCTCCAGT | |
| FHM-IFN-QF | AAAACTCAAATGTGGACGTA | |
| FHM-IFN-QR | GATAGTTTCCACCCATTTCCT | |
| FHM-NOD1-QF | CTCAACGACAGAGGCGAGAG | |
| FHM-NOD1-QR | CGCGTCTTCAGCTTCCTTTG | |
| FHM-TLR3-QF | CCAAACCTGGTGACCCTTGT | |
| FHM-TLR3-QR | AGCCTGGAAACAGCCATTCT | |

(*Continued on next page*)

**TABLE 2** Primers used in this study (*Continued*)

| Name | Sequence (5′–3′) | Remark[a] |
|---|---|---|
| FHM-TRIF-QF | CTCTCCAGAAGCGAGCCAAA | |
| FHM-TRIF-QR | GTCTCGTAGTCTTTGCGCCT | |
| FHM-PKR-QF | CGCTATTACACCGCTTGGGA | |
| FHM-PKR-QR | CTCCTCTCTGGGAAACGCTC | |
| FHM-NF-κB-QF | CTGCTGGAGGGTAACGCATA | |
| FHM-NF-κB-QR | TTTGGTTGAGTCCCGTCCTG | |
| NNV-CP-QF | GGATTTGGACGTGCGACCAA | |
| NNV-CP-QR | CGAGTCAACACGGGTGAAGA | |
| TFV-MCP-QF | TCGCTGGTGGAGCCCTGGTA | |
| TFV-MCP-QR | GGCGTTGGTCAGTCTGCCGTA | |

[a]The special use of primers was indicated. The mutations mediated and the tag added by primers were specified. The accession numbers of each gene were also indicated. A total of nine new nucleotide sequences were uploaded to GenBank.

synthesized with PrimeScript RT reagent kit (TAKARA, Japan). For RT-PCR assay, 20-ng cDNA from cells was used as the template, and the amplified bands were separated by agarose gel electrophoresis. For RT-qPCR of indicated mRNA expression and qPCR of TFV genome copy, the quantitative real-time PCR was conducted by a Light Cycler480 (Roche) instrument (using the 384-well module) with SYBR green master mix (TAKARA, Japan), and the procedure was described as followed: 95°C for 5 min; 95°C for 30 s, 60°C for 30 s, 72°C for 15 s (45 cycles). β-Actin was used as the reference gene. The mRNA relative expression changes and DNA copy of TFV genome were calculated by $2^{-\Delta\Delta Ct}$ method. The primers used were shown in Table 2.

## RNA interference

For gene knockdown, siRNAs targeting RIG-I, MDA5, MAVS, and NOD1 of FHM cells were designed using siDirect version 2.0 online tool (http://sidirect2.rnai.jp/) and were synthesized by RiboBio Co., Ltd. (Guangzhou, China). For each target, there were at least three siRNAs synthesized, and the sequences of siRNAs with higher silencing efficiency were provided in Table 3. The negative control siRNA was set as targeting the *gfp* sequence. FHM cells ($1 \times 10^5$) were co-transfected with 60-pmol siRNA and 1-µg ProA expression vector for RT-qPCR or 60-pmol siRNA and 1-µg ProA expression vector plus pIFN (100 ng) and pRLcmv (20 ng) for dual-luciferase reporter assay and seeded into 24-well plates. After 72 hpt, cells were harvested for RNA extraction followed by RT-qPCR and for dual-luciferase reporter assay.

## Exosome extraction

CCO cells ($1.5 \times 10^7$) were transfected with 20-µg EV or ProA expression vector. At 48 hpt, the supernatants (about 15 mL) were collected, centrifuged at 300 g, and stored 2-mL supernatant for exosome positive (exosome+) sample. The exosome+ supernatant was further used to remove the exosome by ultracentrifugation (70) to obtain an exosome negative (exosome−) sample. Five hundred microliters of exosome+ or exosome− were used to incubate with fresh CCO cells for 24 h. Subsequently, TFV infection (MOI = 2) was carried out, and crystal violet staining was performed at 3 dpi.

## Isolation of nuclear and cytoplasmic extract and immunoblotting

The nuclear and cytoplasmic fractions of FHM cells were prepared using NE-PER Nuclear and Cytoplasmic Extraction Reagents (Thermo Fisher) according to the manufacturer's instructions. Briefly, the treated cells were washed twice with cold PBS and centrifuged at $300 \times g$ for 3 min. Add 200-µL ice-cold CER I to the cell pellet. Vortex the tube vigorously for 15 s to fully suspend the pellet and incubate the tube on ice for 10 min. Add 11-µL ice-cold CER II to the tube. Vortex the tube for 5 s and incubate the tube on ice for 1 min. Vortex the tube for 5 s, and centrifuge for 5 min at $16,000 \times g$. The supernatant

**TABLE 3** Sequences of siRNAs used in this study

| Name | Sequence (5′–3′) |
|------|------------------|
| FHM-RIG-Ii | UUGUUUUUUGGGUUUUUGGGG |
| | CCAAAAACCCAAAAAACAAGG |
| FHM-MDA5i | UAUAAACAGCUUUCUAAAC |
| | GUUUAGAAAGCUGUUUAUAGU |
| FHM-MAVSi | UGUAGUUUCCAGAAGUUUCUC |
| | GAAACUUCUGGAAACUACACU |
| FHM-NOD1i | AUAAUCAUUGAUGUUAUUG |
| | CAAUAACAUCAAUGAUUAUGG |

was transferred to a pre-chilled tube to be the cytoplasmic fraction. The insoluble pellet fraction was resuspended in 100 µL of ice-cold NER by vortexing for 15 s and incubated on ice for 10 min and then centrifuged for 10 min at 16,000 × $g$. The resulting supernatant, constituting the nuclear fraction, was used for the subsequent immunoblotting to detect the changes in the protein level of phosphorylated TBK1 (pTBK1) and IRF3. The internal control of whole cell lysate and cytoplasmic fraction was set to β-actin, while that of nuclear fraction was set to histone H3. The following antibodies were used for immunoblot analysis: anti-pTBK1 rabbit monoclonal antibody (1:1,000; 5483S, Cell Signaling), homemade anti-grouper IRF rabbit polyclonal antibody (1:1,000), anti-β-actin mouse monoclonal antibody (1:20,000; 66009-1-Ig, Proteintech), anti-Histone H3 rabbit monoclonal antibody (1:2,000; 4499s, Cell Signaling), horse anti-mouse IgG-HRP (1:5,000; 7076, Cell Signaling), and goat anti-rabbit IgG-HRP (1:5,000; 7074, Cell Signaling).

## Statistics and reproducibility

Statistical analyses were carried out using Microsoft Excel software and GraphPad Prism. Data were presented as the mean ± SD. Two-tailed unpaired Student's $t$-test was used to calculate the statistical differences between two groups with a confidence interval of 95%, and $P \leq 0.05$ was considered to be statistically significant. All experiments were performed three or more times independently under identical or similar conditions.

## ACKNOWLEDGMENTS

This work was supported by the National Natural Science Foundation of China (31972825, 32273174), Earmarked Fund for China Agriculture Research System (CARS-46), Natural Science Foundation of Guangdong Province (2021A1515011265, 2023A1515012365), and Key Research & Development Program of Guangdong Province (2019B020217001).

S. Huang performed all plasmid construction, dual-luciferase reporter assays, microscopy, RT-qPCR, RNAi, immunoblotting, and all cell-related experiments including cell culture, cytotoxicity assay, IFA, virus propagation, and titering except otherwise specified. T. Su and Y. Huang performed experiments related to MAVS, TRAF3, and TLR3. R. Huang carried out TFV infection and the anti-TFV activity of ProA in CCO cells. L. Su performed all bioinformatic analyses. Y. Wu performed the luciferase activity in different cells. S. Weng supervised the experiments and acquired funding. J. He supervised the work. J. Xie designed the study, guided and supervised the experiments, wrote this manuscript, and acquired funding. All authors read and approved the final manuscript.

## AUTHOR AFFILIATIONS

[1]State Key Laboratory of Biocontrol, Southern Marine Science and Engineering Guangdong Laboratory (Zhuhai), China-ASEAN Belt and Road Joint Laboratory on Mariculture Technology, Guangdong Provincial Key Laboratory of Aquatic Economic Animals, Sun Yat-sen University, Guangzhou, China

2School of Life Science, Huizhou University, Huizhou, China

## AUTHOR ORCIDs

Siyou Huang http://orcid.org/0009-0006-2965-8991
Jianguo He http://orcid.org/0000-0003-2460-7061
Junfeng Xie http://orcid.org/0000-0002-0669-5339

## FUNDING

| Funder | Grant(s) | Author(s) |
|---|---|---|
| China Agricultural Research System (CARS) | 46 | Shaoping Weng |
| MOST | National Natural Science Foundation of China (NSFC) | 31972825, 32273174 | Junfeng Xie |
| GDSTC | Natural Science Foundation of Guangdong Province (廣東省自然科學基金) | 2021A1515011265, 2023A1515012365 | Junfeng Xie |
| Key Research and Development Program of Guangdong Province | 2019B020217001 | Junfeng Xie |

## AUTHOR CONTRIBUTIONS

Siyou Huang, Data curation, Formal analysis, Investigation, Methodology, Validation, Writing – review and editing | Yi Huang, Data curation, Formal analysis, Investigation, Methodology, Validation | Taowen Su, Formal analysis, Investigation, Methodology, Software, Validation, Visualization | Runqing Huang, Conceptualization, Formal analysis, Investigation | Lianpan Su, Formal analysis, Investigation, Methodology, Software, Validation, Visualization | Yujia Wu, Data curation, Formal analysis, Investigation, Methodology | Shaoping Weng, Funding acquisition, Project administration, Resources, Supervision | Jianguo He, Funding acquisition, Project administration, Supervision | Junfeng Xie, Conceptualization, Funding acquisition, Project administration, Resources, Supervision, Writing – original draft, Writing – review and editing

## DATA AVAILABILITY

The sequences of *Channa striata interferon* (MG799351), *RIG-I* (OP805912), *MDA5* (OP805911), *LGP2* (OP805909), *MAVS* (OP805910), *TRAF3* (OP805914), *TBK1* (OP846095), *IRF3* (OP846094), and *RIKP2* (OP805913) were cloned from transcriptome data (unpublished data) and verified by RT-PCR. The sequences of *Pimephales promelas LGP2* (OP805915), *MDA5* (MG799354), *TRIF* (MG799355), *NOD1* (MG799353), and *PKR* (MG799352) were retrieved from the reported genome. The accession numbers of other related genes cloned from *Pimephales promelas* are listed in Table 2.

## ADDITIONAL FILES

The following material is available online.

Supplemental Material

**Fig. S1 (Spectrum04532-22-s0001.tif).** Overexpression of ProA in 293T cells cannot induce human IFN expression.
**Fig. S2 (Spectrum04532-22-s0002.tif).** CCO is an ideal cell line for the study of ProA-mediated IFN activation.
**Fig. S3 (Spectrum04532-22-s0003.tif).** RNAi efficiency in FHM cells.
**Fig. S4 (Spectrum04532-22-s0004.tif).** FHM pTBK1 could be detected by the commercial antibody.
**Supplemental legends (Spectrum04532-22-s0005.docx).** Legends for Fig. S1 to S4.

## Open Peer Review

**PEER REVIEW HISTORY (review-history.pdf).** An accounting of the reviewer comments and feedback.

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
