## [Reviewer comments · Microbiology Spectrum]

Microbiology Spectrum

Orange-spotted grouper nervous necrosis virus-encoded Protein A induces interferon expression via RIG-I/MDA5-MAVS-TBK1-IRF3 signaling in fish cells

Siyou Huang, Yi Huang, Taowen Su, Runqing Huang, Lianpan Su, Yujia Wu, Shaoping Weng, Jianguo He, and Junfeng Xie

Corresponding Author(s): Junfeng Xie, Sun Yat-Sen University

Review Timeline:

Submission Date:	January 12, 2023
Editorial Decision:	June 27, 2023
Revision Received:	August 16, 2023
Accepted:	November 20, 2023

Editor: Luke Iwanowicz

Reviewer(s): The reviewers have opted to remain anonymous.

Transaction Report:

DOI: <https://doi.org/10.1128/spectrum.04532-22>

June 27, 2023

Prof. Junfeng Xie
Sun Yat-Sen University
State Key Laboratory of Biocontrol, School of Life Sciences
135# Xingang Xi Road
Guangzhou, Guangdong 510275
China

Re: Spectrum04532-22 (Orange-spotted grouper nervous necrosis virus-encoded Protein A induces interferon expression via RIG-I/MDA5-MAVS-TBK1-IRF3 signaling in fish cells)

Dear Prof. Junfeng Xie:

I have now received reviewer comments of this manuscript. This research investigates a topic of relevance for journal readership. The reviewers have provided helpful comments that must be addressed in full. In general, these comments request clarification and elaboration. In addition to addressing these comments it would be useful to consider how the virus replicates and virus-cell interaction in addition to cellular responses. Also, as pointed out by Reviewer #1 please elaborate on molecular responses of CCO cells that support viral replication. Contrasting responses of these cell lines is of great value here. Lastly, both grammar and syntax should be revised in the discussion.

Link Not Available

Sincerely,

Luke Iwanowicz

Journals Department
Reviewer comments:

Reviewer #1 (Comments for the Author):

The manuscript provides interesting information on interferon production against NNV infection in susceptible and non-

susceptible cell lines. The study is well designed and appropriate methodology has been used. However, not wanting to make light of the authors' explanation about the need of the RLR signal pathway for IFN stimulation in FHM cells (which do not support NNV replication), I think that more discussion should be focused on the differential response in CCO cells (which does support NNV growth).

Some more detailed comments that I think can help to improve the manuscript.

Introduction

Line 86. Please provide the full name of the virus as is the first time is cited in the text. In addition, redaction should include that NNV belongs to the G. Betanodavirus.

Line 87 Please delete " a kind of"

Line 100 cell lines are susceptible or not to a virus

Results

Line 26. This sentence is a deduction from the obtained results. It fits better in discussion.

Line 155. The authors should provide the TCID50 values obtained after each treatment.

Line 181. The authors may choose to state that the titer was calculated by TCID50 method or not to state the method.

Line 206 Please delete "medium"

Line 233-234 and figure 3H. Although it is true that both mutations have a strong effect on IFN expression I would not say that MDA5 is essential because IFN promoter activity is detected. Please revise this statement

Line 269. No DN overexpression is observed in figure 4D, I see a down-regulation when MAVS-DN is used

Lines 300. The authors should indicate that up-regulation in CCO is much lower than in FHM. In fact, I'm quite surprised that so low differences in mRNA expression in TBK1 expression in CCO are considered significant.

Lines 316-317. The signal pathway is clear in FHM cells, but I do not see so clearly the TBK1 in CCO cells.

Discussion

As a general comment, I recommend reminding briefly which mutants are being discussed . In addition, I miss more detailed discussion about CCO results

Lines 456-458. This sentence is difficult to understand, please re-write .

Line 458-460 "The RLR signal pathway...overexpression of the key factors" Please re-redact this sentence. The experimental support of the need of the RLR signal pathway is the knockdown or reduction the gene expression when siRNAs or negative mutants are used.

Line 506-510. Please re-write this sentence. It is too long and contains some grammatical errors

Line 511-513. To support viral growth a cell must be susceptible (receptors that allow viral attachment are displayed on its surface) and permissive (it allows the completion of the viral cycle and the release of viral progeny) to the virus.

Line 516. I do not understand this last assumption. Why do the authors think that NNV RdRp uses as template the host DNA? All single stranded viruses as NNV form double stranded intermediates during its replication process that I guess could be recognized by RIG-like receptors

M&M

Lines 528-29 full name of SB and ZF4 cells should be provided.

Figure legends

The complete names of the mutants should be provided to make easier interpretation of the results shown in them.

Fig 8. line 432. In this figure no DNs results are shown

Reviewer #2 (Comments for the Author):

Major comments for authors:

Huang et al. performed this study to demonstrate orange-spotted grouper nervous necrosis virus-encoded protein A induced IFN via RIG-I/MDA-5 signaling in fish cells. Though the topic is interesting, the reviewer has some questions on statistical analysis and experimental design. Though the PAMPs for proA were determined, the key regulators are still not found. The study is kind of superficial. Some detailed comments can be found below:

Fig. 1A, "the universal characteristic IFN activation among fish cells" is an overstatement as some fish cell lines just showed less than 2 fold increase. < 2 fold increase should not be statistically significant but the authors indicated they are ($P = 10^{-5}$). Please explain.

Fig. 1B, did the authors try high dose of ProA?

Fig. 3A, RIG-I just caused 1.5 fold increase in mRNA expression. How come they showed $P=0.0001$? Did the authors perform ANOVA before doing two group comparison?

Fig 5 is not a surprising finding as RIG or MDA activation may trigger TBK1 phosphorylation and IRF3 activation.

Fig. 7: ProA can promote TLR3-regulated IFN activation. But why did the authors stated that TLR3 is irrelevant to ProA-mediated IFN activation?

Minor comments for authors:

None

Staff Comments:

Preparing Revision Guidelines

Please return the manuscript within 60 days; if you cannot complete the modification within this time period, please contact me. If you do not wish to modify the manuscript and prefer to submit it to another journal, please notify me of your decision immediately so that the manuscript may be formally withdrawn from consideration by Microbiology Spectrum.

Reviewer(s)' Comments to Author:

Reviewer #1 (Comments to the Author)

The manuscript provides interesting information on interferon production against NNV infection in susceptible and non-susceptible cell lines. The study is well designed and appropriate methodology has been used. However, not wanting to make light of the authors' explanation about the need of the RLR signal pathway for IFN stimulation in FHM cells (which do not support NNV replication), I think that more discussion should be focused on the differential response in CCO cells (which does support NNV growth).

Response: Thank you very much for the valuable comments and suggestions. The revised manuscript has been improved substantially with your help. More discussion about IFN response in CCO cells was added to enrich the CCO part. (Line 522-542 in the “Marked-Up Manuscript”)

The source idea of this manuscript was motivated by the results of ProA-induced IFN promoter with huge differences in FHM and CCO cells. To reveal why the IFN response to ProA was so weak in CCO (the cells support NNV growth), we used FHM (the cells do not support NNV growth but exhibit strong IFN response to ProA) as basic cells and performed comparative analysis to derive the immunity of CCO cells based on the results of FHM cells.

Some more detailed comments that I think can help to improve the manuscript.

1. Introduction

(1) Line 86. Please provide the full name of the virus as is the first time is cited in the text. In addition, redaction should include that NNV belongs to the G. Betanodavirus.

Response: We apologize for the unclear abbreviation. In this revised manuscript, the full name and taxonomy of NNV were added and the sentence has been rephrased as “Nervous necrosis virus (NNV) belongs to the family Nodaviridae genus *Betanodavirus* and is one of the most harmful fish viruses, leading to almost 100% mortalities in juveniles and larvae when infected.” (Line 87-88 in the “Marked-Up Manuscript”)

(2) Line 87 Please delete " a kind of"

Response: Thanks for the suggestion. The phrase “a kind of” was deleted. (Line 89 in the “Marked-Up Manuscript”)

(3) Line 100 cell lines are susceptible or not to a virus

Response: Thank you for the suggestion. We are so sorry for the confusion in our previous draft. Actually, NNV can enter both FHM and CCO cells, suggesting that these two cell lines are susceptible to NNV.

However, NNV cannot replicate in FHM and only a low level of CP expression can be detected in RIG-I-(Δ 1-217) transfected FHM. Therefore, only CCO but not FHM is permissive to NNV. Following the suggestion, all the “sensitive” have been amended to “permissive” in the abstract (Line 31, 33 in the “Marked-Up Manuscript”), in the text (Line 101, 102, 205, 270, 271, 467, 510 in the “Marked-Up Manuscript”), and in the figure legends (Line 1010 in the “Marked-Up Manuscript”).

2. Results

(4) Line 126. This sentence is a deduction from the obtained results. It fits better in discussion.

Response: Thank you for the suggestion. This sentence was remended to “Furthermore, ProA-triggered IFN promoter activation may be specific for fish cells.” and moved to the discussion section. (Line 460 in the “Marked-Up Manuscript”)

(5) Line 155. The authors should provide the TCID₅₀ values obtained after each treatment.

Response: Thank you for the suggestion. All the TCID₅₀ data of TFV was listed in Table 1, including the titer from Fig. S2E, and from Fig2A & Fig2D. (Please refer to the “Table.docx” file)

(6) Line 181. The authors may choose to state that the titer was calculated by TCID50 method or not to state the method.

Response: Thank you for the suggestion. We have deleted “by titer assay” in the figure legend of Fig.S2. (Line 1019 in the “Marked-Up Manuscript”)

In the uploaded R1 version, Fig.S2 and its legend were removed from the main manuscript and added to the supplemental material file.

(7) Line 206 Please delete "medium"

Response: Thanks for the suggestion. We have deleted “medium” in the text. (Line 205 in the “Marked-Up Manuscript”)

(8) Line 233-234 and figure 3H. Although it is true that both mutations have a strong effect on IFN expression I would not say that MDA5 is essential because IFN promoter activity is detected. Please revise this statement.

Response: Thanks for the comment and suggestion. We have changed “essential” to “important”. (Line 232 in the “Marked-Up Manuscript”)

(9) Line 269. No DN overexpression is observed in figure 4D, I see a down-regulation when MAVS-DN is used.

Response: Thanks for the suggestion. We have changed “DN

overexpression” to “transfection of MAVS mutant, MAVS-(Δ 1-95)”.
(Line 266 in the “Marked-Up Manuscript”)

(10) Lines 300. The authors should indicate that up-regulation in CCO is much lower than in FHM. In fact, I'm quite surprised that so low differences in mRNA expression in TBK1 expression in CCO are considered significant.

Response: Thanks for the suggestion. We have added the sentence “Notably, the extent of upregulation in CCO cells is significantly lower compared to FHM cells (Fig. 5A)”. (Line 298 in the “Marked-Up Manuscript”)

As for the second question, we found a statistically significant difference in mRNA expression in TBK1 expression in CCO compared to the control group, although this difference may not be biologically significant. The original data and the *P* values ($P = 0.0007$) calculated by two-tailed unpaired Student's *t*-test were shown below.

		=TTEST(D2:D6,E2:E7,2,2)					
	C	D	E	F	G	H	I
1		EV	ProA	EV	IRF3	EV	IFN
2	TBK1 expression	1.055798	1.24545	1.012136	1.202823	1.0051436	1.617015
3		0.9715319	1.153474	0.948246	1.345678	1.0177982	1.674039
4		1.02498	1.367619	1.055798	1.345678	0.96825	1.32256
5		0.984525	1.258466	0.9715319	1.235101	1.055798	1.80667
6		0.98246	1.1689588	1.055798	1.327152	0.948246	1.628263
7			1.1255989				
9	AVG	1.003859	1.219928	1.008702	1.291286	0.999047	1.609709
10	TTEST		0.00069		6.3E-05		7E-05

(11) Lines 316-317. The signal pathway is clear in FHM cells, but I do not see so clearly the TBK1 in CCO cells.

Response: Thanks for the comment. The CCO cells were deleted in this sentence. (Line 315 in the “Marked-Up Manuscript”)

3. Discussion

As a general comment, I recommend reminding briefly which mutants are being discussed. In addition, I miss more detailed discussion about CCO results.

(12) Lines 456-458. This sentence is difficult to understand, please re-write.

Response: Thanks for the suggestion. The sentence has been revised to “When it is overexpressed in the host cell but not in the state of viral infection, RdRp also exerts RNA replication activity to generate

double-stranded intermediates that cause an RLR-related IFN response. Here, our experimental results show that ProA, the RdRp encoded by OGNNV, induced IFN response by its RNA replication activity.” (Line 455-460 in the “Marked-Up Manuscript”)

(13) Line 458-460 "The RLR signal pathway...overexpression of the key factors" Please re-redact this sentence. The experimental support of the need of the RLR signal pathway is the knockdown or reduction the gene expression when siRNAs or negative mutants are used.

Response: Thanks for the suggestion. We rewrote the sentence and changed it to “Silencing essential components of the RLR signaling pathway using siRNA or negative mutants confirmed that ProA-induced IFN activation relies on this pathway, RIG-I/MDA5-MAVS-TRAF3-TBK1-IRF3.” (Line 460-463 in the “Marked-Up Manuscript”)

(14) Line 506-510. Please re-write this sentence. It is too long and contains some grammatical errors.

Response: Thanks for the suggestion. The sentence has been changed to “In this study, we found that ProA-mediated factors (RIG-I, MDA5, TBK1, IRF3, and IFN) were downregulated in CCO cells during authentic NNV infection, while the negative regulators (LGP2 and

TRAF3) maintained their expression.” (Line 535-538 in the “Marked-Up Manuscript”)

To describe the inhibitory mechanism, a new paragraph was added and merged with the above sentence to discuss two types of virus-caused inhibitory mechanisms targeting IFN signaling. The reason why no or very low IFN response during authentic NNV infection is that there are IFN-inhibiting proteins encoded by NNV, such as B1 (Fish Shellfish Immunol, 2023, 134:108578), B2 (Aquaculture, 2021, 536:736; submitting) and CP (Zool Res, 2022, 43:98-110; J Immunol, 2022, 209:326-336; submitting), or negative regulators of the host cells, such as heat shock protein (submitting), to suppress the RLR signal pathway. (Line 531-542 in the “Marked-Up Manuscript”)

(15) Line 511-513. To support viral growth a cell must be susceptible (receptors that allow viral attachment are displayed on its surface) and permissive (it allows the completion of the viral cycle and the release of viral progeny) to the virus.

Response: Thanks for your comment. We cannot agree more. Following your opinion, the sentence has been revised and extended to a single paragraph.

“In general, a cell line that supports productive viral infection must be susceptible and permissive to the virus. For susceptible cells, they contain

the receptors that allow viral attachment and entry, exhibiting the ability to support the absorption of viruses. Both FHM and CCO cells in this study are susceptible to NNV. For permissive cells, they allow the completion of the viral cycle and the release of viral progeny, possibly as they have several abnormalities in their antiviral intrinsic immunity factors or restriction factors for viral traffic. For instance, the CCO cells that are weak in IFN response could well support NNV replication while FHM only supported a low level of NNV replication when RIG-I and MDA5 were partially silencing. Therefore, we meditate that the level of IFN response to ProA is one of the determinants for the successful replication of NNV.” (Line 522-530 in the “Marked-Up Manuscript”)

(16) Line 516. I do not understand this last assumption. Why do the authors think that NNV RdRp uses as template the host DNA? All single stranded viruses as NNV form double stranded intermediates during its replication process that I guess could be recognized by RIG-like receptors
Response: Thanks for the comment and opinion. The original assumption is specified for the present study that the overexpressed ProA induces IFN response and not for the state of authentic NNV infection. You are right that NNV generates double-stranded intermediates during its replication and RLRs do recognize the intermediates. According to the suggestion, We have revised the assumption to “Based on our previous

results (30) and the current data, we put forward a hypothesis that ProA produces dsRNA during NNV replication, and the synthesized intermediates are mainly detected by RIG-I-like receptors (RIG-I and MDA5), resulting in IFN transcription through the essential signaling of RIG-I/MDA5-MAVS-TRAF3-TBK1-IRF3. To suppress the IFN response, negative regulation must be taken by NNV via modulating host factors or expressing viral protein. We identified that LGP2, NOD1, and PKR play a negative regulatory role in ProA-mediated IFN transcription at the RNA sensor level.” (Line 543-549 in the “Marked-Up Manuscript”)

4. M&M and Figure legends

(17) Lines 528-29 full name of SB and ZF4 cells should be provided.

Response: Thanks for the comment and suggestion. We have provided the full names of SB and ZF4 cells. SB refers to Asian sea bass cells, while ZF4 cells represent the zebrafish embryonic fibroblast cell line. The information has been corrected in the Marked-Up Manuscript. (Line 120-121; Line 556-557 in the “Marked-Up Manuscript”)

(18) The complete names of the mutants should be provided to make easier interpretation of the results shown in them.

Response: Thanks for the comment and suggestion. We have provided the

complete names of the dominant negative (DN) mutants. RIG-I-DN, MDA5-DN, MAVS-DN, TRAF3-DN1, TRAF3-DN2, and TRIF-DN have been changed to RIG-I-(Δ 1-217), MDA5-(Δ 1-284), MAVS-(Δ 1-95), TRAF3-(C71A/H73A), TRAF3-(Δ 425-548), and TRIF-(302-459). The information about DN has all been corrected. (Line 34, 41; Line 229-230, 266, 272, 273, 377, 378, 404, 466,498, 577-583, 923-924, 937, 940, 979, 988 in the “Marked-Up Manuscript”) The primers used for DN construction were also renamed in Table 2. (Please refer to the “Table.docx” file)

(19) Fig 8. Line 432. In this figure no DNs results are shown

Response: Thanks for the suggestion. The figure legend of Fig 8E has been corrected as “CP expression representing NNV replication was evaluated by RT-qPCR in FHM cells transfected with RIG-I, MDA5, EV, RIG-I-(Δ 1-217) and MDA5-(Δ 1-284) at different infection time points as indicated.” (Line 987-989 in the “Marked-Up Manuscript”).

Reviewer #2 (Comments to the Author)

Major comments for authors:

Huang et al. performed this study to demonstrate orange-spotted grouper nervous necrosis virus-encoded protein A induced IFN via RIG-I/MDA-5 signaling in fish cells. Though the topic is interesting, the reviewer has

some questions on statistical analysis and experimental design. Though the PAMPs for ProA were determined, the key regulators are still not found. The study is kind of superficial.

Response: Thank you very much for your review and comments. The revised manuscript has been improved substantially with your help. In this manuscript, we found that overexpression of ProA can activate IFN response in fish cells. Combining the results of the previous work (Fish Shellfish Immunol, 2018, 79:234-243), we deduced that ProA generates dsRNA with its RdRp activity, dsRNA as PAMP is recognized by RLR, the RLR signal pathway is activated, and IFN is expressed finally. For the methods of statistical analysis and part of the raw data please see the detailed responses below.

Some detailed comments can be found below:

1. Fig. 1A, "the universal characteristic IFN activation among fish cells" is an overstatement as some fish cell lines just showed less than 2 fold increase. < 2 fold increase should not be statistically significant but the authors indicated they are ($P = 10^{-5}$). Please explain.

Response: Thanks for the comment. In all figures, data were presented as the mean \pm standard deviation. Two-tailed, unpaired Student's t-test was used to calculate the statistical differences (P values) between the two groups. Biostatistical analysis revealed a statistically significant

difference, although this difference may generally consider not to be biologically significant. However, dose-dependent experiments revealed that ProA induces species-specific IFN promoter activation in a dose-dependent manner, suggesting that although the elevated fold is less than 2, it is biologically significant. An Excel file containing the original data of Fig.1 was uploaded (Please refer to the “Fig1 Data-upload.xlsx” file).

2. Fig. 1B, did the authors try high dose of ProA?

Response: Thank you for your comment. We did perform the transfection of all four NNV proteins, CP, B1, B2, and ProA, into 293T at various concentrations to test the expression in preparation for interaction and bulk expression purification. However, only B2 can modulate the IFN response in 293T (suppression). We did not observe activation from a high dose of ProA in 293T. Therefore, ProA is proven to not activate IFN response in 293T. Furthermore, we used hMAVS as a positive control and observed significant activation of ISRE upon transfection with the same plasmid. Interestingly, when using the same dose of ProA, it was able to induce the expression of IFN in different fish cells.

3. Fig. 3A, RIG-I just caused 1.5 fold increase in mRNA expression.

How come they showed $P=0.0001$? Did the authors perform ANOVA

before doing two group comparison?

Response: Thank you for your comment and suggestion. The experiment was independently replicated three times, and the resulting *P* value was calculated by GraphPad using a two-tailed unpaired Student's *t*-test. The result was found to be 0.0001. We conducted a reanalysis of our data using one-way ANOVA, which revealed significant differences among the groups as shown in the following snapshot.

Ordinary one-way ANOVA									
Multiple comparisons									
1	Number of families	1							
2	Number of comparisons per family	5							
3	Alpha	0.05							
4									
5	Dunnett's multiple comparisons test	Mean Diff.	95.00% CI of diff.	Below threshold?	Summary	Adjusted P Value	A-?		
6	EV vs. RIG-I	-0.4366	-0.6234 to -0.2498	Yes	****	<0.0001	B	RIG-I	
7	EV vs. EV	-0.001772	-0.1866 to 0.1850	No	ns	>0.9999	C	EV	
8	EV vs. MDA5	-0.1321	-0.3280 to 0.06381	No	ns	0.2781	D	MDA5	
9	EV vs. EV	0.000	-0.1868 to 0.1868	No	ns	>0.9999	E	EV	
10	EV vs. LGP2	-0.4317	-0.6405 to -0.2228	Yes	****	<0.0001	F	LGP2	
11									
12	Test details	Mean 1	Mean 2	Mean Diff.	SE of diff.	n1	n2	q	DF
13	EV vs. RIG-I	0.9946	1.431	-0.4366	0.06962	6	6	6.271	27
14	EV vs. EV	0.9946	0.9984	-0.001772	0.06962	6	6	0.02546	27
15	EV vs. MDA5	0.9946	1.127	-0.1321	0.07301	6	5	1.909	27
16	EV vs. EV	0.9946	0.9946	0.000	0.06962	6	6	0.000	27
17	EV vs. LGP2	0.9946	1.426	-0.4317	0.07783	6	4	5.546	27
18									

4. Fig 5 is not a surprising finding as RIG or MDA5 activation may trigger TBK1 phosphorylation and IRF3 activation.

Response: Thank you for your comment. We agree with your opinion. In our previous study, we demonstrated that ProA is capable of activating IFN through RLR pathways and depends on TBK1 and IRF3. In this manuscript, we extended our findings and studied the TBK1 phosphorylation and IRF3 translocation, identifying the essential involvement of TBK1 and IRF3 in ProA-mediated IFN activation. There are different branches of the RLR signal pathway in different species and from different stimulants of various PAMP. We confirmed that

ProA-induced IFN activation relies on this pathway, RIG-I/MDA5-MAVS-TRAF3-TBK1-IRF3. It is worthwhile to characterize the detailed RLR pathway and revealed the mechanism of the interaction between NNV proteins and host innate immune signaling.

5. Fig. 7: ProA can promote TLR3-regulated IFN activation. But why did the authors stated that TLR3 is irrelevant to ProA-mediated IFN activation?

Response: Thanks for the suggestion. We are sorry for the confusion. The results depicted in Figure 7A-B demonstrate the ability of ProA to upregulate the expression of TLR3 and TRIF (its crucial downstream junction protein), indicating that TLR3 and TRIF are ISGs. Furthermore, Figure 7C reveals that the overexpression of TRIF-(302-459), the dominant negative mutant of TRIF, does not impact ProA-induced IFN activation. Therefore, we provisionally conclude that TLR3 is not involved in ProA-mediated IFN activation. The previous results (Fish Shellfish Immunol, 2018, 79:234-243) revealed that ProA expression did not upregulate cytokine factors including TNF α , TGF β , and IL-10. Consequently, the signal transduction of ProA-mediated IFN response does not go through the TLR-TRIF-NF- κ B branch.

Re: Spectrum04532-22R1 (Orange-spotted grouper nervous necrosis virus-encoded Protein A induces interferon expression via RIG-I/MDA5-MAVS-TBK1-IRF3 signaling in fish cells)

Dear Prof. Junfeng Xie:

The authors have adequately address reviewer comments. Please carefully review the manuscript during the proofing stages to address a few remaining typos and word omissions.

Your manuscript has been accepted, and I am forwarding it to the ASM production staff for publication. Your paper will first be checked to make sure all elements meet the technical requirements. ASM staff will contact you if anything needs to be revised before copyediting and production can begin. Otherwise, you will be notified when your proofs are ready to be viewed.

Sincerely,
Luke Iwanowicz
Editor
Microbiology Spectrum